# Early-life exposure to low-dose oxidants can increase longevity via microbiome remodelling in *Drosophila*

Fumiaki Obata [1,2], Clara O. Fons[1] & Alex P. Gould [1]

Environmental stresses experienced during development exert many long-term effects upon health and disease. For example, chemical oxidants or genetic perturbations that induce low levels of reactive oxygen species can extend lifespan in several species. In some cases, the beneficial effects of low-dose oxidants are attributed to adaptive protective mechanisms such as mitohormesis, which involve long-term increases in the expression of stress response genes. Here we show in *Drosophila* that transient exposure to low concentrations of oxidants during development leads to an extension of adult lifespan. Surprisingly, this depends upon oxidants acting in an antibiotic-like manner to selectively deplete the microbiome of *Acetobacter* proteobacteria. We demonstrate that the presence of *Acetobacter* species, such as *A. aceti*, in the indigenous microbiota increases age-related gut dysfunction and shortens lifespan. This study demonstrates that low-dose oxidant exposure during early life can extend lifespan via microbiome remodelling rather than mitohormesis.

---

[1] Physiology and Metabolism Laboratory, The Francis Crick Institute, 1 Midland Road, London, NW1 1AT, UK. [2] Present address: Department of Genetics, Graduate School of Pharmaceutical Sciences, The University of Tokyo, 7-3-1 Hongo, Bunkyo-ku, Tokyo, 113-0033, Japan. Correspondence and requests for materials should be addressed to A.P.G. (email: Alex.Gould@crick.ac.uk)

Nutrition, stress, toxins and other environmental factors experienced during development (early life) can have long-term effects upon adult health and disease[1–4]. Early-life environmental factors such as nutrition also impact upon lifespan and, depending upon the condition, they can either shorten or prolong it[4–6]. The combined evidence from human epidemiology and rodent models for the developmental origins of health and disease (DOHaD) is now overwhelming. However, the processes by which the early-life environment exerts long-term influences upon adult physiology are only just emerging. These are often termed programming mechanisms and are likely to include epigenetic regulation of gene expression, permanent structural changes to organs and altered cellular ageing[7–10].

Mild increases in superoxide or other reactive oxygen species (ROS) are known to produce beneficial effects via the induction of mitohormesis and related adaptive stress responses[11–13]. Mitohormesis involves the long-term upregulation of genes encoding antioxidant enzymes, the mitochondrial unfolded protein response (UPR$^{mt}$) and other factors mediating adaptive stress responses thought to increase lifespan. In *Caenorhabditis elegans*, exposure to a low dose of the oxidant paraquat (*N*,*N*'-dimethyl-4,4'-bipyridinium dichloride) during both development and adulthood is known to extend longevity[14]. Extended lifespan in *C. elegans* can also be triggered by partial loss of function of components of the mitochondrial electron transport chain (ETC)[14,15]. ETC perturbations and low-dose paraquat can both lead to a mild increase in mitochondrial superoxide, which is thought to act as a signal inducing protective mechanisms that attenuate the effects of ageing[14]. Likewise, in mice and in *Drosophila melanogaster*, decreased expression of ETC genes is also associated with extended lifespan[16,17]. In *Drosophila*, muscle-specific genetic perturbation of the ETC promotes longevity via a mitohormetic effect involving the UPR$^{mt}$ and *Impl2*, a negative regulator of insulin signalling[18]. Here we develop *Drosophila* DOHaD models for transient early-life exposure to low doses of oxidants and show that they produce increases in lifespan, as well as long-term changes in adult lipid metabolism. Surprisingly, however, the underlying longevity mechanism cannot be accounted for by mitohormesis. Instead, we find that it depends upon oxidant-induced remodelling of the early-life microbiome.

## Results

### Larval exposure to low-dose oxidants extends lifespan.
To identify early-life environmental factors that promote longevity in *D. melanogaster*, an inbred isogenic strain ($w^{iso31}$) was exposed to sublethal doses of environmental stressors during larval development but not during adulthood (Fig. 1a). *Tert*-butyl hydroperoxide (tBH) is known to increase ROS and oxidative stress in *Drosophila* larvae[19] and we found that a concentration of 20 mM was toxic, decreasing larval growth and viability, as well as substantially shortening lifespan (Supplementary Fig. 1a–c). Nevertheless, larval exposure to doses ranging from 1.25 to 10 mM extended the median and maximum lifespan of both males and females by up to 30% (Fig. 1b, c and Supplementary Table 1). Intriguingly, the effect of larval tBH on adult survival does not display a clear dose–response relationship but, instead, appears to switch on an all-or-none longevity response. The longevity response to tBH is not restricted to a $w^{iso31}$ genetic background and it is observed in Canton S and heterogeneous outbred *white Dahomey* ($w^{Dah}$) flies (Fig. 1d, e and Supplementary Figure 1d, e). In females, longevity correlated with decreased fecundity, suggesting that it could be at the cost of reproduction (Supplementary Fig. 1f). Importantly, the longevity response can also be triggered by larval exposure to a low dose (1 mM) of another oxidant, paraquat, which has a very different molecular structure

from tBH (Fig. 1f). Together, the survival analysis shows that exposure to low doses of oxidants during development acts via a long-term mechanism to extend the lifespan of both sexes in a range of different genetic backgrounds.

Adults derived from tBH- or paraquat-exposed larvae (hereafter called tBH- or paraquat-experienced flies) are not only long-lived but they also have altered lipid metabolism. Exposure to either oxidant during larval life led to an ~2-fold increase in the triglyceride content of adult flies at 1 week of age (Fig. 1g and Supplementary Fig. 1g, h). Further investigation of tBH-experienced flies showed that a larger triglyceride store correlates with increased starvation resistance, although the magnitude of resistance in females is less pronounced than that of males and it is also strain-dependent (Supplementary Fig. 2a–f). The results thus far indicate that exposure to low-dose oxidants during development promotes adult triglyceride storage, starvation resistance and longevity.

### tBH selectively removes *Acetobacter* from the microbiome.
To identify the mechanism mediating the long-term effects of low-dose oxidants, we first tested whether or not it might involve mitohormesis. We observed that a low-dose of tBH (5 mM) was sufficient to induce, in larvae, the expression of genes involved in the oxidative stress response (*gstd2* and *sod2*), the UPR$^{mt}$ (*hsp60* and *hsc70-5*) and insulin inhibition (*Impl2*) (Fig. 2a). Interestingly, these genes were all upregulated in the tissue that is directly exposed to dietary tBH, the larval gut, but not in a more internal tissue that plays adipose/liver-like roles, the larval fat body[20]. Moreover, after treated animals had metamorphosed into tBH-experienced adult flies, we did not detect any significant increase in gut or whole-body expression of the panel of oxidative stress response, UPR$^{mt}$ or insulin inhibitor genes (Fig. 2b, c). This strongly suggests that tBH does not trigger a long-term adaptive stress response with the previously described features of mitohormesis. Consistent with the lack of a long-term adaptive response, tBH-experienced flies did not show increased resistance when challenged with tBH or with paraquat during adulthood (Fig. 2d, e). Therefore, in our *Drosophila* DOHaD model, early-life exposure to oxidants is unlikely to promote longevity via a long-term mitohormetic response.

Clues to an alternative longevity mechanism came from the observation of abundant DAPI$^+$ rod-shaped microbes in the lumen of guts from tBH-experienced but not control animals (Supplementary Fig. 3a). These microbes could be eliminated with a broad-spectrum antibiotic cocktail, indicating that they are bacteria (Supplementary Fig. 3b). The microbiome of *D. melanogaster* is less complex than that of mammals— dominated by several species of Gram-positive *Lactobacillaceae* and Gram-negative *Acetobacteraceae*[21–23]. It is known to provide beneficial effects upon larval growth, fecundity and immunity[24–30]. However, the microbiome can also induce age-related gut pathology and, depending upon the context, can either shorten, extend or have no major effect upon lifespan[31–37]. In our laboratory, the larval and adult gut microbiomes of $w^{iso31}$ flies are dominated by lactobacilli, including *Lactobacillus plantarum*, and also by *Acetobacter aceti* (Fig. 3a, original data deposited at DDBJ with accession number DRA005828). Analysis of beta diversity using Bray-Curtis dissimilarity shows that larval tBH treatment leads to an adult gut microbiome that is different from controls, and rarefaction and Shannon alpha diversity measurements suggest that tBH treatment decreases the overall number of bacterial operational taxonomic units (Supplementary Fig. 4a–c). In particular, larval tBH exposure produces a specific and striking change to the gut microbiome of both $w^{iso31}$ larvae and adults: strongly decreasing the relative amount of major (*A. aceti* and

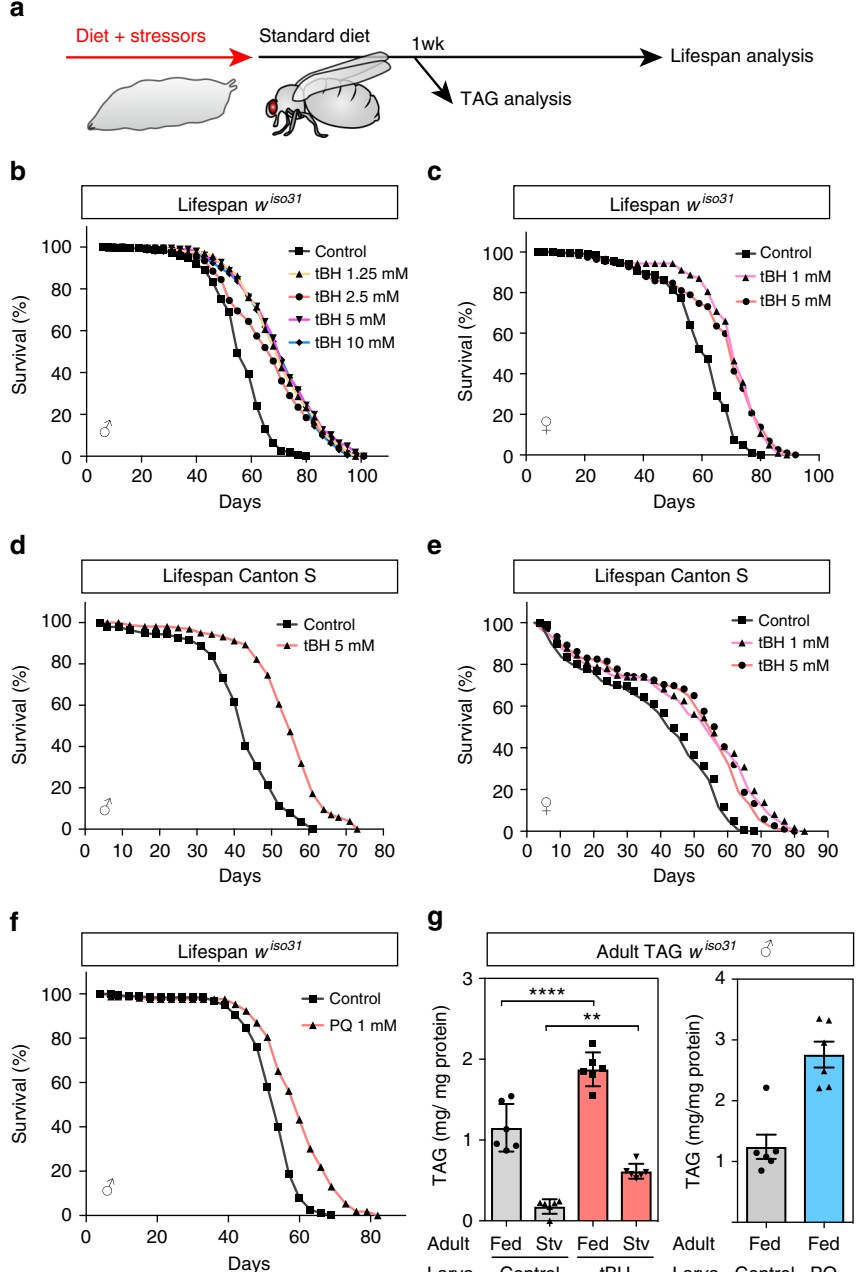

**Fig. 1** Low-dose oxidants during development increase adult fat storage and longevity. **a** Outline of experimental strategy. **b**–**e** Lifespan of $w^{iso31}$ male (**b**, **d**) or female (**c**, **e**) flies, raised on 1–20 mM tBH. **f** Lifespan of $w^{iso31}$ male flies raised on 1 mM paraquat. **g** Whole-body triacylglycerol (TAG) amount in $w^{iso31}$ male flies of 1 week of age, with or without starvation for 24 h. Mean ± SEM ($n = 6$). **p < 0.01 and ****p < 0.0001, see Methods for details of statistical tests used in this and all subsequent figures. Statistics for survival (lifespan) curves are summarised in Supplementary Table 1

*Komagataeibacter rhaeticus*) and minor *Acetobacteraceae* but, importantly, not *Lactobacillus* species (Fig. 3a). Quantitative PCR (qPCR) analysis confirmed that larval tBH eliminates *A. aceti* in both larval and adult guts, and it also revealed that there is a concomitant increase in the absolute numbers of *L. plantarum* in adults (Fig. 3b, c). Similar tBH-induced depletion of *A. aceti* and enrichment of *L. plantarum* was also observed with the somewhat different microbiome of the Canton S strain (Fig. 3a, d). Strikingly, when gut bacteria were cultured on agar plates in the absence of a *Drosophila* host, *A. aceti* but not *L. plantarum* was eliminated by 5 mM tBH (Fig. 3e). Together, these results demonstrate that low-dose tBH acts as an antibiotic that is selective for *Acetobacteraceae* ssp. but not *Lactobaccillus* ssp. This

raises the possibility that oxidants could increase longevity via a selective antibiotic mechanism.

**Antibiotic depletion of *Acetobacter* associates with longevity.** The antibiotic G418, widely used in research, is also reported to kill *Acetobacter* but not *Lactobacillus* spp.[29]. We found that larval treatment with G418, like tBH, depleted the microbiome of *A. aceti* and enriched it for *L. plantarum* (Fig. 4a–c). Moreover, adult triglycerides and longevity were increased similarly in G418- and tBH-experienced flies (Fig. 4d, e). Paraquat treatment of larvae also has a strikingly similar effect on the adult microbiome, as well as mimicking the longevity and triglyceride increases seen with tBH and G418 (Figs. 4c and 1f, g). Hence, early-life exposure

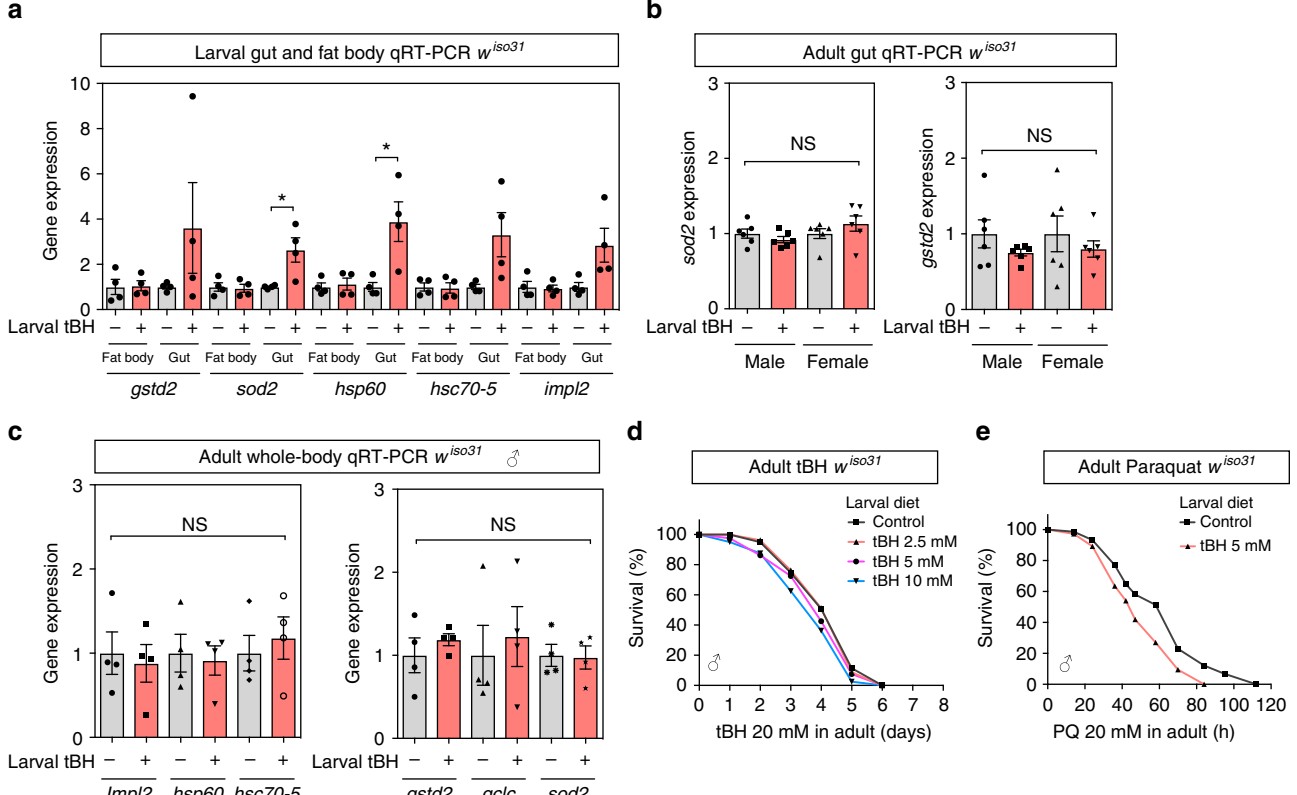

**Fig. 2** Larval tBH exposure induces oxidative stress and mitochondrial unfolded protein responses in larval gut but not in larval fat body or adult body. **a** Quantitative RT-PCR analysis of genes involved in the oxidative stress response (*gstd2* and *sod2*), the mitochondrial unfolded protein response (*hsp60* and *hsc70-5*) and *Impl2* in fat body or gut of $w^{iso31}$ larvae. Mean ± SEM ($n = 4$). **b** Quantitative RT-PCR analysis of *sod2* and *gstd2* in the adult gut of $w^{iso31}$ male or female flies. Mean ± SEM ($n = 6$). **c** qRT-PCR analysis in whole body of $w^{iso31}$ male flies. Mean ± SEM ($n = 4$). **d, e** Survival curves of $w^{iso31}$ male flies exposed to 20 mM tBH (**d**) or 20 mM paraquat (**e**). *$p < 0.05$, NS: not significant. Statistics for survival curves are in Supplementary Table 1

to three very different chemicals that extend longevity is associated with remodelling of the microbiome via the selective depletion of *Acetobacter* but not lactobacilli.

To address whether adult exposure to chemicals remodelling the microbiome could also alter longevity, flies were exposed to tBH or G418 during their first 6 days after eclosion and then transferred to standard diet for the remainder of adulthood (Supplementary Fig. 5a). As with larval exposure, adult treatment with 5 mM tBH initially depleted *A. aceti* and increased *L. plantarum* (Supplementary Fig. 5b). However, after tBH was removed, there was a progressive recolonisation by *A. aceti* and a concomitant decrease of *L. plantarum* (Supplementary Fig. 5b). This transient rather than stable remodelling of the microbiome was associated with an early increase in triglycerides but shortened survival in both tBH- and G418-treated flies (Supplementary Fig. 5c, d). This contrasts with larval tBH exposure, which extends longevity and is associated with a life-long depletion of *A. aceti* from the microbiome (Supplementary Fig. 6a, b). This stable alteration of the microbiome can be passed to the next generation, which live longer than controls despite never being exposed themselves to any exogenous oxidants or antibiotics (Fig. 5a–d). These results together indicate that oxidant/antibiotic exposure is more effective during early life than adulthood at stably remodelling the microbiome and extending lifespan.

## Low-dose oxidants extend lifespan by depleting *Acetobacter*.
The results thus far raise the possibility that either the enrichment of lactobacilli or the depletion of *Acetobacteraceae* from the normal microbiome could be the causal mechanism that increases

triglycerides and longevity. It has been reported that monoassociation of germ-free *Drosophila* with some *Acetobacter* species, including *A. aceti*, can decrease adult triglyceride storage[38,39]. We now find that adult treatment with broadspectrum antibiotics increases triglycerides, starvation resistance and longevity in a manner that cannot be further augmented by tBH (Fig. 6a–e). In fact, tBH-experienced flies are even slightly longer lived when treated as adults with broad-spectrum antibiotic cocktails (Fig. 6d, e). These findings are consistent with the hypothesis that *Acetobacteraceae* may be rate limiting for longevity but, if they are removed from the microbiome, then lactobacilli and/or other tBH-resistant bacteria become mildly limiting.

To test directly the hypothesis that selective loss of *A. aceti* from the normal microbiome is the longevity mechanism, we developed microbiome complementation assays. These take advantage of clonal isolates of *A. aceti* and *L. plantarum* derived from single colonies of $w^{iso31}$ gut microbiota (designated *A. aceti* FO1 and *L. plantarum* FO3). Association of young adults with *L. plantarum* FO3 does not significantly decrease the numbers of *A. aceti* (Fig. 7a–c). However, *A. aceti* FO1 is able to colonise the gut of tBH-experienced flies in a stable manner and also to decrease strongly the numbers of *L. plantarum* (Fig. 7b, c). Thus, in the context of the complex microbiome of $w^{iso31}$ flies, *A. aceti* can suppress or outcompete *L. plantarum* but not vice versa. We also observed that reassociation of young adult flies with *A. aceti* FO1 but not *L. plantarum* FO3 was sufficient to abrogate the increase in triglycerides of tBH-experienced flies (Fig. 7d). Reassociation with *A. aceti* FO1 does not, however, significantly alter triglycerides or lifespan in control flies that were not treated

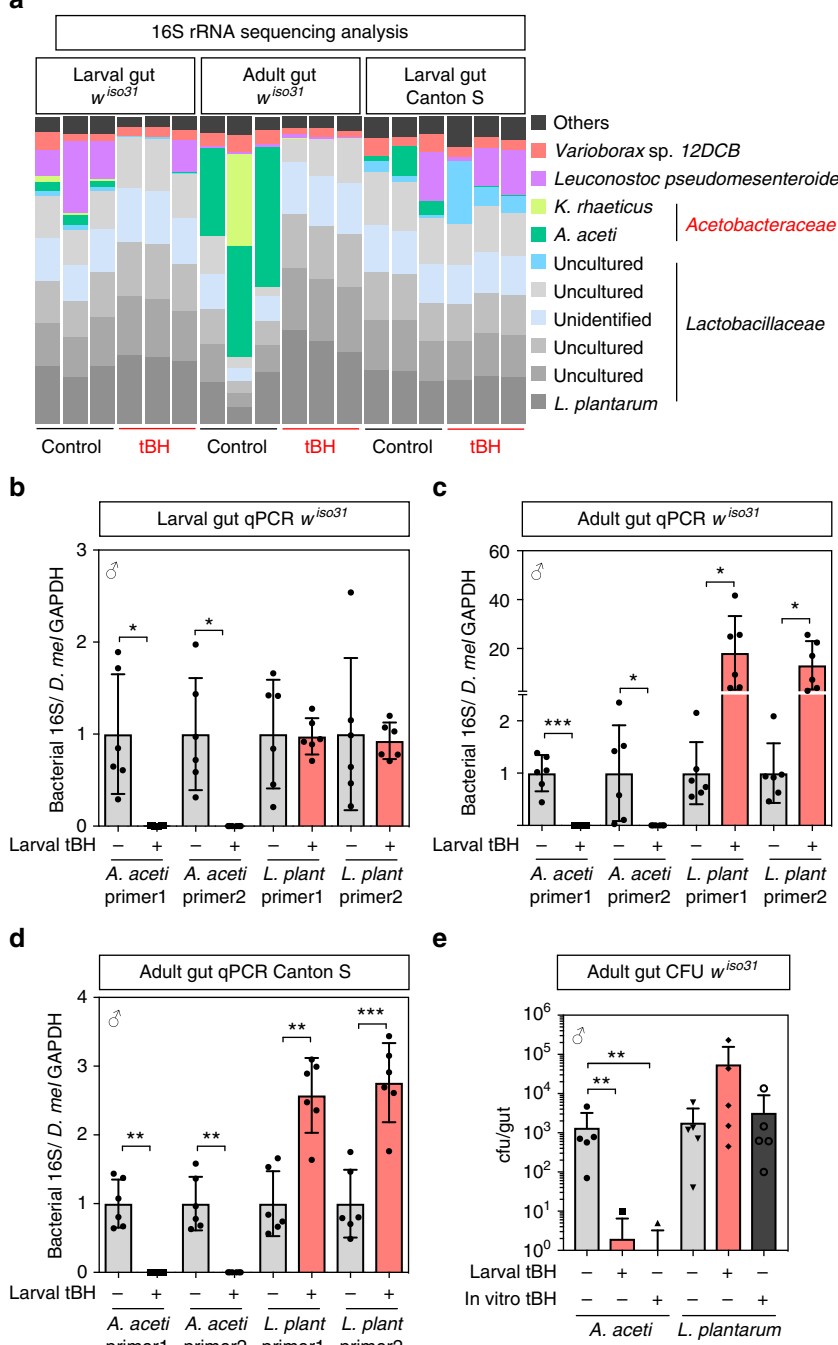

**Fig. 3** tBH selectively depletes *Acetobacteraceae* from the microbiome. **a** 16S metagenomic analysis from conventional control (black) or tBH-treated (red) larval gut (left) or adult gut (middle) from $w^{iso31}$ flies or larval gut from Canton S flies (right). Each column represents a biological replicate and colours indicate operational taxonomic units (OTUs). **b**–**d** Quantitative PCR with species-specific primers for bacteria in larval (**b**) or adult (**c**) guts of $w^{iso31}$ and Canton S (**d**) male flies. Mean ± SEM ($n = 6$). **e** Quantification of colony-forming units (cfu) in the gut from control (grey) or tBH-experienced (red) $w^{iso31}$ male flies. Control gut samples were also plated on 5 mM tBH-containing agar (in vitro tBH+). Mean ± SEM ($n = 5$). *$p < 0.05$, **$p < 0.01$, ***$p < 0.001$

with oxidant (Fig. 7d–f). Strikingly, however, *A. aceti* FO1 completely abolished the lifespan extension of both tBH- and paraquat-experienced flies (Fig. 7e, f). These microbiome complementation assays demonstrate that the presence of a single *Acetobacter* species, *A. aceti*, in the indigenous microbiome can not only decrease triglycerides but also shorten lifespan.

We next asked whether *Acetobacter* species other than *A. aceti* FO1 also shorten lifespan. To test this, we isolated another *Acetobacter* clone from our $w^{iso31}$ flies. This is designated *Acetobacter* FO2 and it is most closely related to *Acetobacter*

*pomorum* and *Acetobacter pasteurianus* (16S rDNA sequence has 99.7% identity to both species). As with *A. aceti* FO1, association of paraquat-experienced flies with *Acetobacter* FO2 reversed the lifespan extension (Fig. 7f). This demonstrates that lifespan-shortening can be induced by different *Acetobacter* species and so could be a property of many members of this bacterial genus. Together, our results thus far show that developmental exposure to low-dose oxidants extends lifespan via a mechanism that is dependent upon the selective removal of *Acetobacter* species from the microbiome.

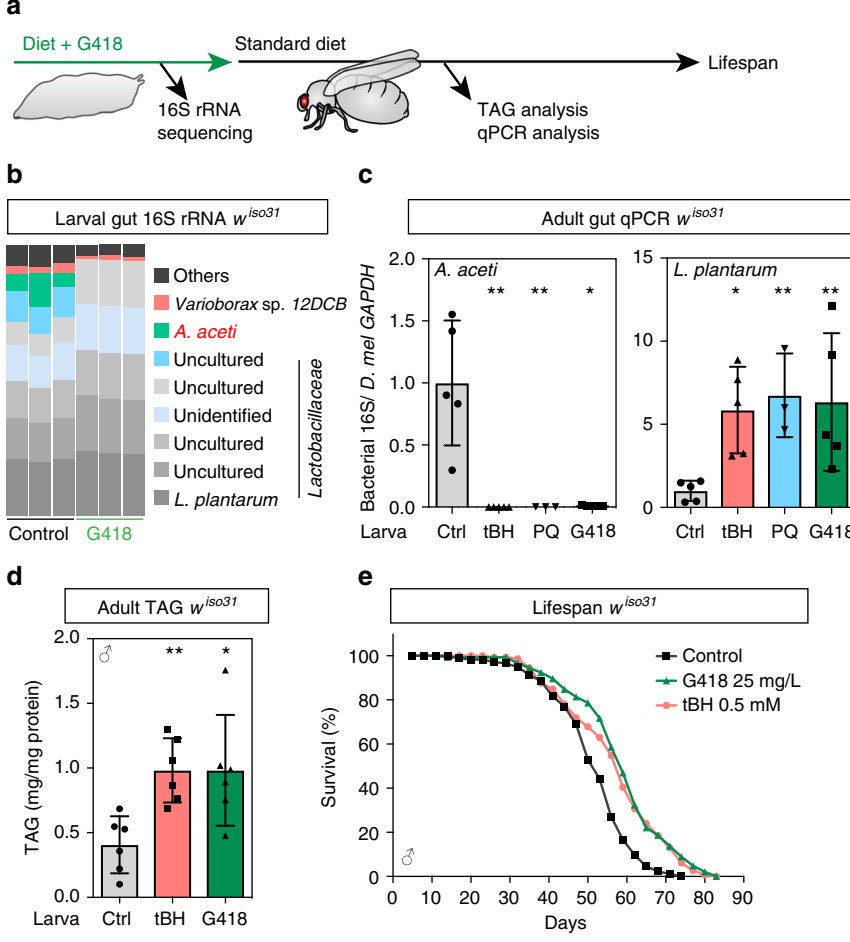

**Fig. 4** *Acetobacter* decreases TAG stores and shortens lifespan. **a** Experimental strategy for G418 treatment. **b** 16S rRNA-sequencing analysis of G418 (25 mg/L)-treated $w^{iso31}$ larval guts. Each column represents a biological replicate and colours indicate OTUs. **c** Quantitative PCR using species-specific primers of bacteria in $w^{iso31}$ male flies raised on tBH (0.5 mM, red), paraquat (PQ; 1 mM, blue) or G418 (25 mg/L, green). Mean ± SEM ($n = 5$ except PQ ($n = 3$)). **d, e** Whole-body TAG (**d**) or lifespan (**e**) of $w^{iso31}$ male flies raised on tBH (0.5 mM, red) or G418 (25 mg/L, green). Mean ± SEM ($n = 6$). *$p < 0.05$, **$p < 0.01$. Statistics for survival curves are shown in Supplementary Table 1

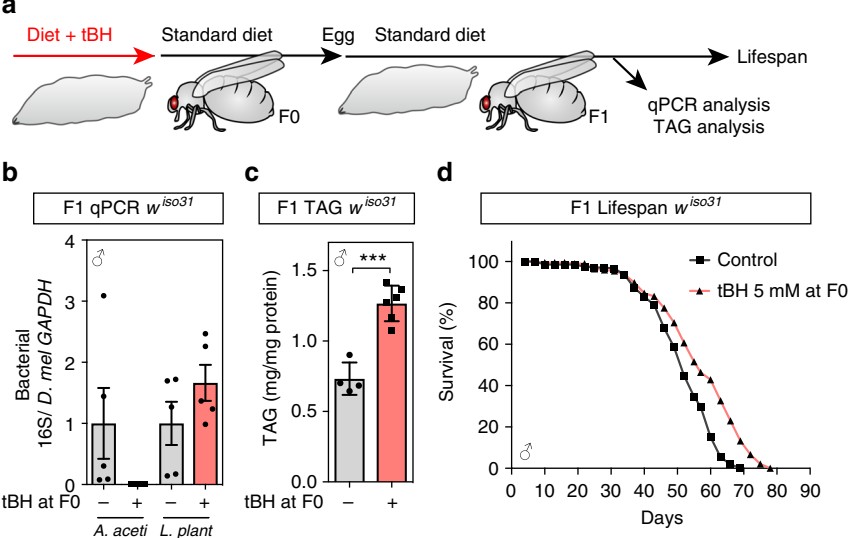

**Fig. 5** Transmission of the tBH microbiome, TAG storage and longevity to the next generation. **a** Outline of experimental strategy. Male and female tBH-experienced flies (F0) were mated for 0–5 days after eclosion, transferred to a new bottle and eggs collected for 4 h to establish the next generation (F1). **b** qPCR analysis of bacteria from F1 male flies using species-specific primers. Mean ± SEM ($n = 5$). **c** Whole-body TAG (mg/mg protein) in F1 male flies. Mean ± SEM ($n = 4$ for control and 6 for tBH). **d** Lifespan of F1 male flies. ***$p < 0.001$. Statistics for survival curves are shown in Supplementary Table 1

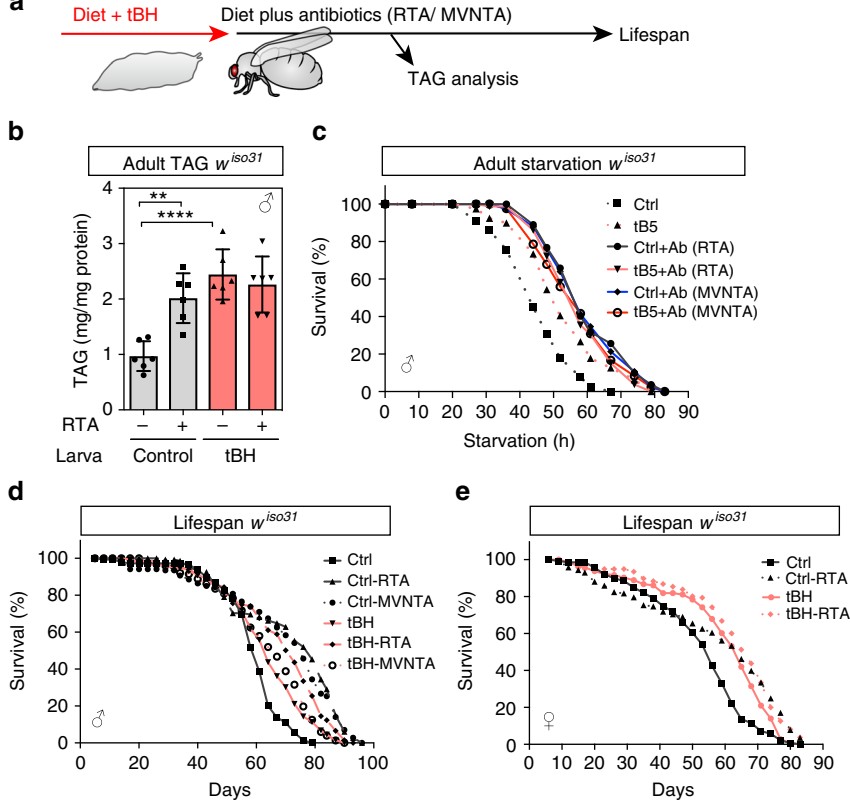

**Fig. 6** Adult-onset antibiotics increase TAG, starvation resistance and longevity. **a** Outline of experimental strategy. **b** Whole-body TAG (mg/mg protein) of control and tBH-experienced male $w^{iso31}$ flies treated as adults for 6 days with rifamycin, tetracycline and ampicillin (RTA). Mean ± SEM ($n = 6$). **c** Starvation survival curves of male $w^{iso31}$ flies on PBS/agar medium with or without RTA or metronidazole, vancomycin, neomycin, tetracycline and ampicillin (MVNTA). **d**, **e** Lifespan of male (**d**) or female (**e**) control or tBH-experienced $w^{iso31}$ flies on standard diet with or without RTA or MVNTA antibiotics. $**p < 0.01$, $****p < 0.0001$. Statistics for survival curves are shown in Supplementary Table 1

**Removal of *Acetobacter* prolongs intestinal healthspan.** To address how *A. aceti* may shorten lifespan, we examined how the bacterial load changes during ageing. Total gut bacterial loads greatly increased between 1 and 6 weeks of age in tBH-experienced and control flies of both sexes (Supplementary Fig. 6a, b). At 4–6 weeks of age, guts from tBH-experienced animals had almost no detectable *A. aceti* but considerably more *L. plantarum* than those from controls. Longevity is known to be limited in female flies by an age-related breakdown of intestinal immune homeostasis and intestinal stem cell (ISC) hyperproliferation, attributable to increasing activity of the FoxO and immune deficiency (IMD) pathways[31,32,40,41]. We found that age-related changes in the expression of FoxO target genes (*InR* and *PGRP-sc2*) in the gut are similar in both control and tBH-experienced flies (Supplementary Fig. 7). Nevertheless, tBH-experienced flies show a striking suppression of the age-related increase in gut and whole-body expression of IMD target genes encoding the antimicrobial peptides diptericin, drosocin and attacin A (Fig. 8a, b). Age-related hyperactivation of IMD targets is suppressed in tBH-experienced flies despite the presence of a higher overall bacterial load than in controls (Fig. 8a, b, Supplementary Fig. 6a, b). Furthermore, adult add back of the *A. aceti* FO1 clone to tBH-experienced flies is sufficient to restore gut IMD hyperactivation in aged flies (Fig. 8c). Importantly, age-related ISC overproliferation is decreased in tBH-experienced female flies, indicating a prolongation of gut healthspan, and this can be reversed by adding back *A. aceti* FO1 (Fig. 8d). These findings together identify *A. aceti* as an indigenous component of the $w^{iso31}$ microbiome that can act as a potent driver of IMD hyperactivation and dysfunction in the gut during ageing.

## Discussion

This study identifies a mechanism linking early-life oxidant exposure to longevity. A key finding is that low-dose oxidants selectively deplete *Acetobacter* from the microbiome during development and, in turn, this can ameliorate age-related gut dysfunction and extend host lifespan. However, *Acetobacter* have beneficial as well as harmful effects upon their *Drosophila* host, promoting growth and reproduction[24,29,30]. In a natural environment, *Acetobacter* likely confer a 'live fast, die young' lifestyle where the selective advantages of faster larval growth and increased fecundity may outweigh any disadvantages of a shortened lifespan. This selective pressure could then explain why *Acetobacteraceae* are a ubiquitous keystone component of the microbiomes of wild caught *Drosophila* from diverse habitats.

We found that early-life but not adult exposure to low-dose oxidants could deplete *Acetobacter* in a long-lasting manner and so extend longevity, even into the next generation. Interestingly, a next-generation effect of G418 treatment delaying larval development has also been attributed to efficient long-term *Acetobacter* depletion[29]. Our observation that dietary supplementation with oxidants can more effectively remove *Acetobacter* from larvae than from adults may be connected with their different feeding strategies or host gut environments. Either way, transient exposure of young adults to oxidants initially depleted the vast majority of *A. aceti*, as judged by qPCR, yet the microbiome was subsequently able to readjust homeostatically. If complete and specific removal of *Acetobacter* during adult stages could be achieved by an alternative method, it remains open as to whether or not this would extend longevity. What is, however, clear from our microbiome complementation experiments is that

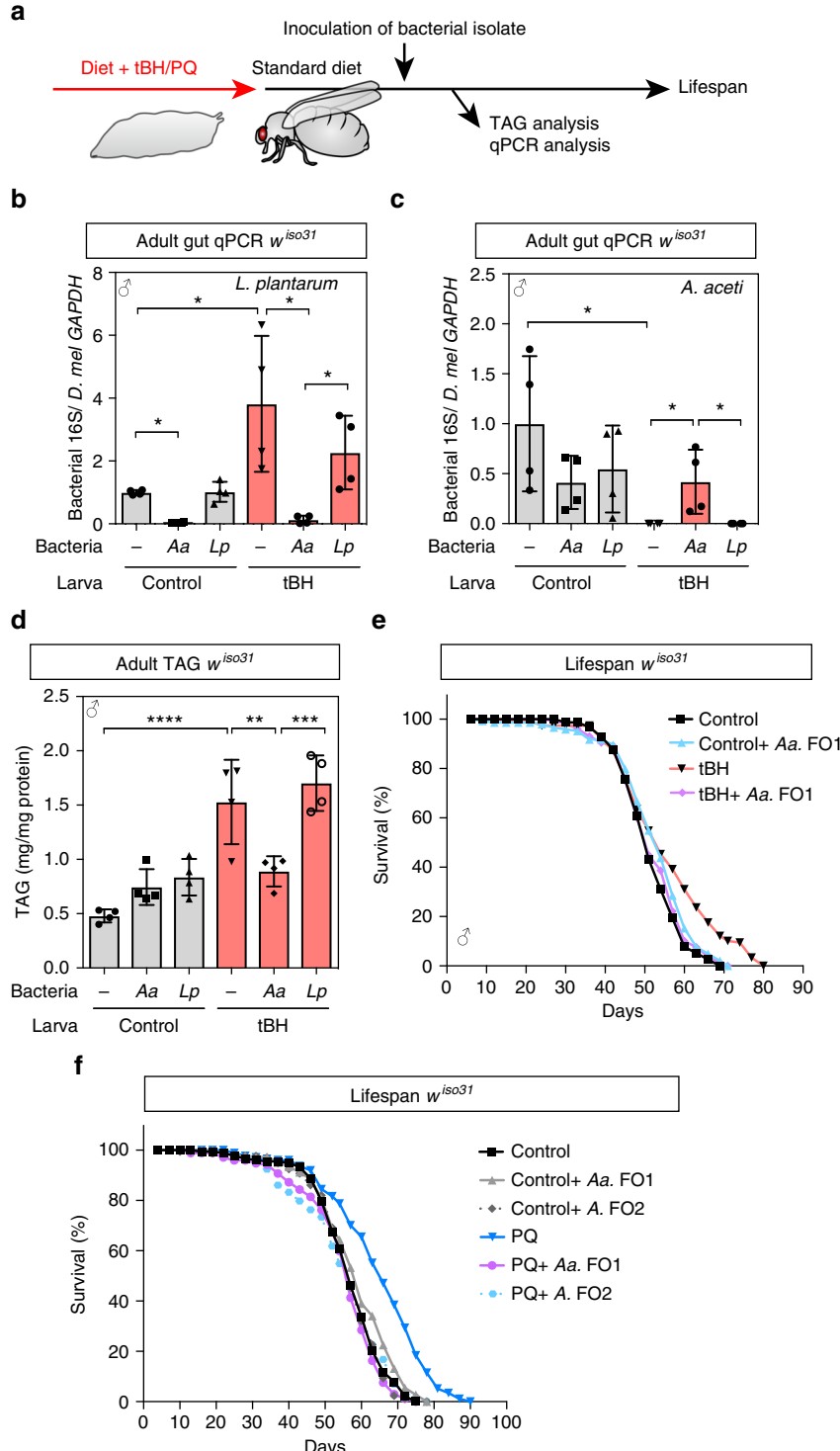

**Fig. 7** *Acetobacter* decreases TAG stores and shortens lifespan. **a** Experimental strategy for adult inoculation of bacterial isolates to tBH-experienced flies. **b**, **c** Quantitative PCR of bacteria in $w^{iso31}$ male flies raised as larvae on control (grey) or tBH (5 mM, red) diets and inoculated with MRS medium only (−), *A. aceti* FO1 (*Aa*) or *L. plantarum* FO3 (*Lp*). Mean ± SEM (*n* = 4). **d**, **e** Whole-body TAG (**d**) and lifespan (**e**) of inoculated $w^{iso31}$ male flies raised on 5 mM tBH. **d** Mean ± SEM (*n* = 4). **f** Lifespan of inoculated $w^{iso31}$ male flies raised on 1 mM PQ. Either *A. aceti* FO1 or *Acetobacter* FO2 is inoculated in adult stage. *$p < 0.05$, **$p < 0.01$, ***$p < 0.001$, ****$p < 0.001$. Statistics for the survival lifespan experiments are shown in Supplementary Table 1

interactions between *Acetobacter* and its host during adult stages are sufficient to induce gut dysfunction and to limit lifespan.

It is surprising that low-dose oxidants extend *Drosophila* lifespan via their action upon microbiota rather than upon host mitohormesis. Oxidant depletion of *Acetobacter* but not lactobacilli is a selective antibiotic-like effect that can occur in a Petri

dish in the absence of any host tissues. This would appear to rule out a critical role for mitohormesis or any other early-life host stress responses during the initial oxidant-induced bacterial selection process. Moreover, the microbiome complementation assays with two different *Acetobacter* species demonstrate that oxidant-induced lifespan extensions are strictly dependent upon

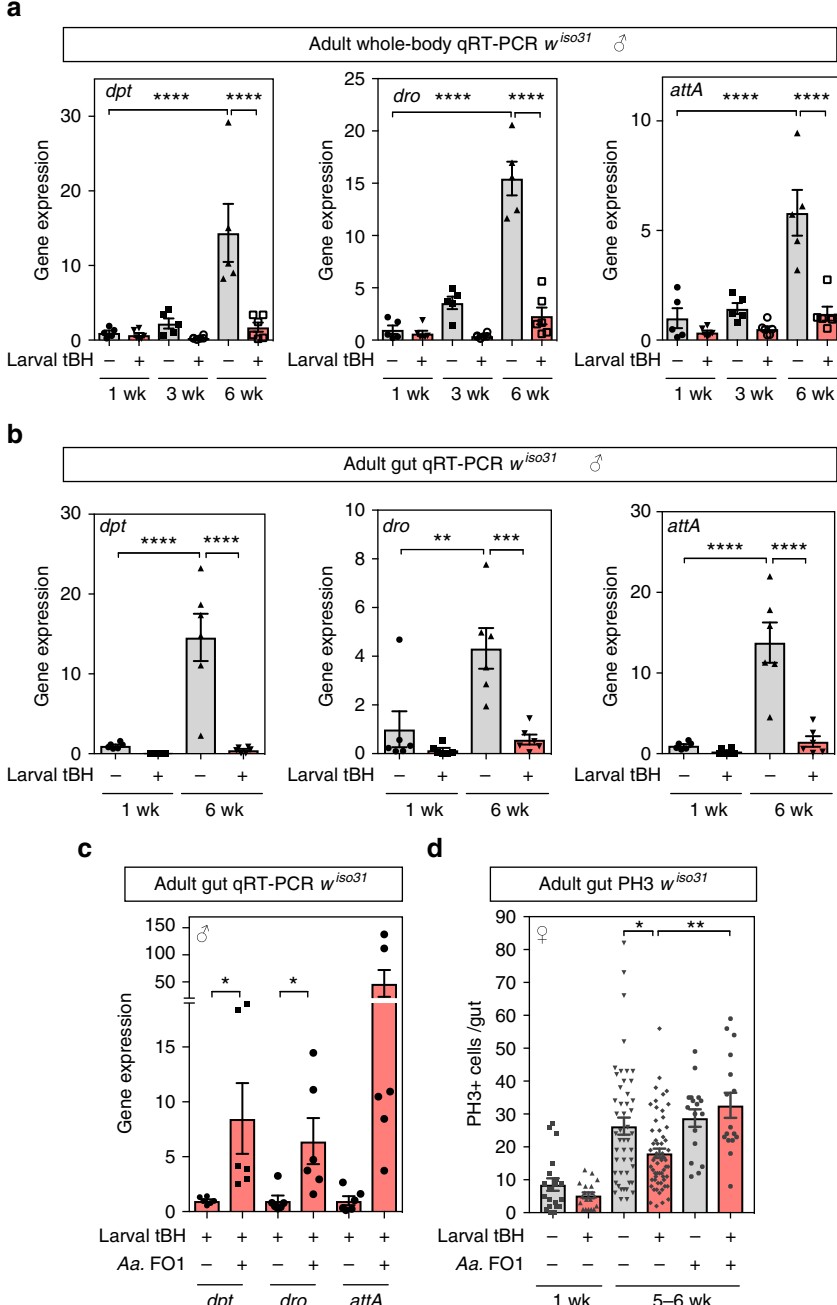

**Fig. 8** Depletion of *A. aceti* suppressed age-related intestinal dysregulation. **a**, **b** Quantitative RT-PCR for three IMD target genes (*dpt*, *dro* and *attA*) in whole body (**a**) or gut (**b**) of *w*^iso31^ male flies at 1, 3 and 6 weeks of age. Mean ± SEM (*n* = 6). **c** Quantitative RT-PCR of the three IMD target genes in guts from tBH-experienced *w*^iso31^ male flies at 5 weeks of age, with or without reassociation with *A. aceti* FO1. Mean ± SEM (*n* = 6). **d** The number of phospho-histone H3-positive (PH3+) cells in the midgut of *w*^iso31^ male flies with or without reassociation with *A. aceti* FO1. Mean ± SEM (from left to right *n* = 20, 19, 48, 59, 17 and 16). *$p < 0.05$, **$p < 0.01$, ***$p < 0.001$, ****$p < 0.0001$

the absence of members of this bacterial genus. Nevertheless, our experiments do not rule out a dual mechanism whereby oxidants remodel the microbiome but they also induce a long-lasting hormetic or other host stress response, such that both effects need to work together to extend lifespan. Therefore, in *C. elegans*, where low-dose paraquat is known to increase longevity via a ROS signal acting on the host[14], it would still be interesting to test whether microbiome remodelling makes a contribution. These experiments may best be done in the context of the nemotode's natural microbiome, which is complex and includes *Acetobacteraceae*, rather than with the *Escherichia coli* monoassociation commonly used in laboratory studies[42].

This *Drosophila* study illustrates that targeted depletion of specific bacteria from the early-life microbiome can provide an efficient method to extend adult healthspan. In humans and other mammals, early-life environmental factors such as diet and antibiotic exposure can remodel the microbiome, in turn, influencing adult physiology, metabolism and behaviour[36,43–50]. Emerging evidence also indicates that mammalian and fish microbiota, like that of flies, may influence healthspan and lifespan[45,51,52]. Future studies will be needed to sift carefully through complex mammalian microbiomes to identify those bacterial species (and their variants) that play functional roles in age-related diseases and in longevity.

## Methods

**Fly husbandry**. *D. melanogaster* stocks were raised on a standard yeast-cornmeal diet containing per litre: 23.4 g autolysed yeast extract (Brian Drewitt); 58.5 g glucose (VWR, Cat. No. 10117HV); 66.3 g cornmeal (Brian Drewitt); 7.02 g agar (Brian Drewitt); and 19.5 mL of antimycotic solution containing 0.04% bavistan (Sigma-Aldrich, Cat. No. 378674) and 10% nipagin (Sigma-Aldrich, Cat. No. H3647). The main wild-type strain used in this study was *white iso31* ($w^{iso31}$)[53]. *Wolbachia* was removed from the original $w^{iso31}$ line using 50 μg/mL tetracycline for four generations. *Wolbachia*-negative $w^{iso31}$ was then maintained on standard diet for more than 10 generations to allow restoration of microbiota. *Wolbachia* was not cleared from the Canton S or from the *white Dahomey* ($w^{Dah}$) strain[54]. Embryos were collected by minimal washing from grape juice-agar plates supplemented with live yeast paste so that parent-derived microbiota could be transferred to the next generation. All flies in this study were raised as larvae under constant density by adding a fixed volume of embryos (12 μL, ~150 embryos) to 30 mL of fly food in each 250 mL polyethylene bottle. Adult flies were collected within 2 days of eclosion and maintained for an additional 2 days on standard diet for maturation and mating, followed by male–female separation under light $CO_2$ anaesthesia. Unless otherwise stated, the separated males or females were then maintained at a density of 25 male or 15 female flies per vial and transferred to fresh vials every 2–3 days for the duration of the experiment.

**Larval exposure to oxidants**. Diets containing tBH (Sigma, 458139) or paraquat (methyl viologen dichloride hydrate, Sigma, 856177) were prepared by adding 50 mL of a 20× stock solution to 950 mL of standard diet after it had cooled down to ~70 °C. In all experiments using tBH diet, the control diet was prepared from the same batch of food by adding 50 mL of reverse osmosis purified (Milli Q) water. Diets were stored at 4 °C for 2 weeks at most.

**Survival analysis and stress resistance assays**. Survival curves were determined by flipping flies into fresh vials every 2–3 days and counting dead flies each time. All flies were maintained under constant temperature (25 °C) and humidity (65%) with a 12 h light–dark cycle. In most cases, seven (for male) or nine (for female) vials, each containing 25 (for male) or 15 (for female) flies, were analysed for each survival curve. For starvation assays, flies were transferred to a vial containing phosphate-buffered saline (PBS)/1% agar, and deaths recorded at intervals as indicated in the graphs. Five vials containing 15 flies each were counted for each condition. For adult oxidative stress survival assays, 20 mM tBH or 20 mM paraquat were prepared in 5% sucrose and PBS/1% agar.

**Larval tBH survival and adult body weight measurements**. A fixed volume of embryos (12 μL) per 30 mL of fly food in a 250 mL polyethylene bottle was used to measure survival on standard diet supplemented with tBH at different concentrations. Survival was calculated from the numbers of larvae that pupariated, relative to the control (0 mM tBH). The body weights of individual adult flies were measured using an ultrasensitive scale (Sartorius, MSE3.6P000DM).

**Triacylglyceride measurements**. Triacylglycerides (TAGs) were quantified using a colorimetric glycerol assay. Five male or female flies were homogenised in 150 μL of 0.1% Triton-X/PBS on ice using a pellet pestle motor (Kontes) with pestle (Sigma, Z359947). After 5 min incubation at 70 °C with shaking at 750 rpm, the samples were centrifuged at 17 000 g for 3 min and supernatants were then collected. To quantify TAG concentrations, 15 μL of the supernatant was incubated for 1 h with an equal volume of triglyceride reagent (Sigma, T2449) or PBS (for negative controls). A volume of 20 μL of the samples was then dispensed into a 96-well plate and incubated with 100 μL of free glycerol reagent (Sigma, F6428). The concentration was calculated based on the serial dilution of a standard (Sigma, G7793). To measure protein concentration, 10 μL of the supernatant was used for BCA assays (Sigma, BCA1) according to the manufacturer's instructions using bovine serum albumin as a standard.

**16S metagenomic analysis and qPCR of bacteria**. Larvae or adults were rinsed in 50% (v/v) bleach (Fisher Chemical, S/5040/PB17), 70% ethanol (Fisher Chemical, E/0650DF/15) and then washed extensively with PBS before dissection. Larval guts (5 per sample) or adult guts (8 per sample) were dissected in sterile PBS using sterile forceps and carefully removing the trachae, malpighian tubules and crop. Guts were collected in PBS on ice, then transferred to 360 μL of lysis buffer (20 mM Tris pH8.0, 2 mM EDTA and 1% Triton X-100) with 20 mg/mL lysozyme from chicken egg (Sigma, L4919) and homogenised using a pellet pestle motor (Kontes) with pestle (Sigma, Z359947). Homogenised samples were stored at −80 °C. Frozen gut samples were thawed at 37 °C for 45 min in a 1.5 mL microcentrifuge tube, transferred to a 2 mL tube (Sarstedt, 72.693.005) containing 0.1 mm glass beads (Scientific Industries, SI-BG01) and then shaken using a Mini-Beadbeater-24 (Biospec Products, 112011EUR) at 2500 rpm for 20 s. After 15 min incubation at 37 °C, 40 μL of proteinase K and 200 μL of Buffer TL (Qiagen) were added to each sample, followed by incubation at 56 °C for 15 min. Genomic DNA was purified by QIAamp DNA Micro kit (Qiagen, 56304) and 16S rRNAs amplified using primers (8F-YM and Bakt 357 R), which target the V1–V2 variable region. PCR amplicons were purified using a QIAquick PCR Purification kit (Qiagen, 28104) and sent to IMGM Laboratories, GmbH, for Illumina MiSeq sequencing. Sample-specific

barcode sequences and sequencing adapters were added to the PCR amplicons to index them. All indexed amplicons were purified using Agencourt AMPure XP beads (Beckman Coulter, A63881), normalised on a SequalPrep normalisation plate (Thermo Fisher Scientific, A1051001) and subsequently pooled into one sequencing library. Integrity of the library was checked on an Bioanalyzer DNA-1000 lab chip (Agilent Technologies, 5067–1504) and its concentration was measured using the Qubit dsDNA HS assay (Thermo Fisher Scientific, Q32854). Sequencing was performed on an MiSeq next-generation sequencing system (Illumina) with its 500 cycle v2-chemistry, generating 2 × 250 bp paired-end reads. The sequencing run performed well with 87.1% of all bases with a Q-score > 30. A high sequencing depth between 294.212 and 531.268 reads per sample was obtained. The phylogenetic analysis was performed with the CLC Genomics Workbench (version 9.5.3; Qiagen) and its Microbial Genomics module (version 1.6.1). Paired-end reads were merged, quality trimmed and primer sequences were removed. The resulting reads were trimmed to a fixed length of 280 bp and operational taxonomic unit (OTU) clustering was performed using the SILVA SSU database v123 with a 99% sequence similarity threshold for the annotation of OTUs. The Canton S strain is *Wolbachia*-positive but these reads accounted for <1% of the total and so they were not removed from the analysis. Low-abundance OTUs with <10 reads over all samples were discarded, resulting in 260 annotated OTUs and 1.097 de novo OTUs, which were not represented in the database. Rarefaction curves, alpha diversity (Shannon entropy) and beta diversity (PCo plots of Bray-Curtis dissimilarity) were calculated according to the guidelines of the CLC Microbial Genomics Module (Qiagen). For the graphs shown in Figs. 3a and 4b, the data were thresholded to remove OTUs below 0.01% of total abundance. Data are deposited at DDBJ with the accession number DRA005828.

For quantification of bacterial species by qPCR, two different species-specific primers were used for *A. aceti*[29,55] and *L. plantarum*[29,56], although some cross-reaction from closely related species cannot be excluded. All values were normalised to the *Drosophila GAPDH* gene and in all graphs, except Fig. 3, primer set1 was used.

Pan-bacterial: 5′-CCTACGGGAGGCAGCAG-3′, 5′-ATTACCGCGGCTGCT GG-3′

*A. aceti* primer set1: 5′-TAGTGGCGGACGGGTGAGTA-3′, 5′- AATCAAA CGCAGGCTCCTCC-3′

*A. aceti* primer set2: 5′-TGGAGCATGTGGTTTAATTCGA-3′, 5′- GCGGGA AATATCCATCTCTGAA-3′

*L. plantarum* primer set1: 5′-CGAACGAACTCTGGTATTGATTG-3′, 5′- ACC ATGCGGTCCAAGTTG-3′

*L. plantarum* primer set2: 5′-AGGTAACGGCTCACCATGGC-3′, 5′- ATTCCC TACTGCTGCCTCCC-3′

*D. melanogaster GAPDH*: 5′-TAAATTCGACTCGACTCACGGT-3′, 5′- CTCC ACCACATACTCGGCTC-3′

**Quantitative reverse transcription-PCR of *Drosophila* gene expression**. Whole bodies (8 flies per sample) or dissected tissues (5 larval tissues per sample or 5–8 adult guts per sample) were homogenised in Qiazol (Qiagen, 79306) using a pellet pestle motor (Kontes) with pestle (Sigma, Z359947). RNA purification was performed using the RNeasy micro kit (Qiagen, 74004). RNA concentrations were estimated using a Nanodrop 2000 (ThermoFischer Scientific) and 200 ng of total RNA per sample were reverse-transcribed using the Superscript IV first-strand synthesis system (Thermo Fisher Scientific, 18091050) with Oligo (dT)$_{20}$ primers. qPCR was performed using a LightCycler 480 with SYBR Green I Master (Roche, 04887352001). The data in the figures show fold change relative to the control after normalisation to *RNA polymerase II*.

*RNA pol II*: 5′-CCTTCAGGAGTACGGCTATCATCT-3′, 5′-CCAGGAAGACC TGAGCATTAATCT-3′

*gstd2*: 5′-CTCCAATGTCTCCAGGTGGT-3′, 5′-CCCAGTTCTCATCCCAT CC-3′

*sod2*: 5′-AATTTCGCAAACTGCAAGC-3′, 5′-TGATGCAGCTCCATGATC TC-3′

*hsp60*: 5′-TGATGCTGATCTCGTCAAGC-3′, 5′-TACTCGGAGGTGGTGT CCTC-3′

*hsc70-5*: 5′-GGAATTGATATCCGCAAGGA-3′, 5′-TCAGCTTCAGGTTCATG TGC-3′

*impl2*: 5′-GCCGATACCTTCGTGTATCC-3′, 5′-TTTCCGTCGTCAATCCAA TAG-3′

*gclc*: 5′-CGAGGAGAATGAGCTGTTCC-3′, 5′-ACCAGACCCGGAAAAAC G-3′

*dpt*: 5′-GTTCACCATTGCCGTCGCCTTAC-3′, 5′-CCCAAGTGCTGTCCAT ATCCTCC-3′

*dro*: 5′-CCATCGAGGATCACCTGACT-3′, 5′-CTTTAGGCGGGCAGAAT G-3′

*attA*: 5′-CACAATGTGGTGGGTCAGG-3′, 5′-GGCACCATGACCAGCATT-3′

*PGRP-sc2*: 5′-CCAAGTCTATCGGCATCTCC-3′, 5′-GAGCAGAGGTGAGG GTGTTG-3′

*InR*: 5′-GCAAACTCTGCCAGACGAA-3′, 5′-CGCATCCACCCAAACAAT-3′

**Colony-forming unit assays**. A single gut was dissected and homogenised using the pellet pestle motor (Kontes) with pestle (Sigma, Z359947) in 100 μL of MRS

medium (Oxoid 10249582). After 1 min centrifuge at 200 rpm, the supernatant was spread onto a semi-selective plate and, following incubation, the number of colonies counted. For selective *Lactobacillus* culture, MRS agar (Oxoid) plates were used at 37 °C with a Gaspak for anaerobic conditions to suppress *Acetobacter* growth. For selective *Acetobacter* culture, mannitol agar (2.25 g/L D-mannitol, 5 g/L yeast extract, 3 g/L peptone and 15 g/L agar) plates was used at 29 °C with aerobic conditions.

**Microbiome complementation assays**. Bacteria were plated from the gut of 1-week-old $w^{iso31}$ male flies using the selective culture methods above and individual colonies were picked and identified as below. Clonal isolates (*A. aceti* FO1, *Acetobacter* FO2 and *L. plantarum* FO3) were cultured to $OD_{600} = 1$ in MRS medium (Oxoid, 10249582) at 29 °C (For FO1 and FO2) or 37 °C (for FO3). The 16S rDNA sequence of FO1 matches more than 99% to *A. aceti*, *Acetobacter sicerae* and *Acetobacter orleanensis*, while that of FO2 to *A. pomorum* and *A. pasteurianus*. The 16S rDNA sequence of FO3 matches 100% to *L. plantarum* and *Lactobacillus pentosus*. For *Acetobacter*, the *Hsp60* gene was also partially sequenced[57]. This Hsp60 analysis indicates that FO1 has 99% identity to *A. aceti* and *A. sicerae*, whereas FO2 has 99% identity to *A. pomorum* and *A. pasteurianus*. Species identifications were further resolved using MALDI ToF-MS (Animal and Plant Health Agency, UK). This indicated that the best match to FO1 is *A. aceti* and to FO3 is *L. plantarum* but *Acetobacter* FO2 was not reliably identified at the species level. For adult inoculations, 40 μL (~ $10^8$ cfu/mL) of culture were added to the surface of a standard food vial and allowed to dry at 29 °C. Control samples were prepared by adding 40 μL of the medium. Day-2 adult flies were transferred without anaesthesia to the bacteria-innoculated food and cultured for 2–4 days.

**Antibiotic treatments**. For the elimination of the microbiome, 1-week-old males were transferred to standard diet containing one of two antibiotic cocktails: RTA (200 μg/mL rifamycin, 50 μg/mL tetracycline and 500 μg/mL ampicillin); or MVNTA (100 μg/mL metronidazole, 50 μg/mL vancomycin, 100 μg/mL neomycin, 50 μg/mL tetracycline and 100 μg/mL ampicillin). Flies were maintained on antibiotic diet continuously for survival analysis or for 6 days for TG and starvation resistance assays. G418 (Sigma, A1720) was dissolved in ethanol (Fisher Chemical, E/0650DF/15) as a 100× stock, and the control for G418 experiments was 1% ethanol in standard diet.

**Immunostaining and confocal microscopy**. For 4′,6-diamidino-2-phenylindole (DAPI) bacterial staining, whole guts were fixed in 4% paraformaldehyde (PFA, Cat. No. 28908, Thermo Scientific) in PBS (Gibco, Life Technologies) for 30 min. Tissues were then washed five times in PBT (PBS with 0.3% Triton X-100) and incubated with DAPI (1 μg/mL; Sigma-Aldrich, Cat. No. D9542) and Alexa Fluor 633-phalloidin (Invitrogen, A22284) in PBT for 15 min at room temperature. Tissues were then rinsed five times in PBT and three times in PBS. Before being mounted in Vectashield (Vector Labs) and imaged using a Leica SP5 confocal microscope. Image processing was performed using Fiji.

For anti-phospho-histone H3 quantifications, whole guts were dissected in PBS, fixed in 4% PFA in PBS for 1 h, followed by permeabilization in PBT (0.3% Triton X-100) for 20 min. Tissues were then incubated in blocking solution (PBS with 10% normal goat serum, G9023, Sigma-Aldrich), and incubated with rat anti-phospho-histone H3 (1/1000, phospho-Ser28, ab10543, Abcam) at 4 °C overnight. Guts were then washed four times with PBT over 1 h, and incubated with Alexa Fluor 488-labelled goat anti-rat antibodies (1/1000, RefA11006, Life Technologies) and DAPI (1 μg/mL; Sigma-Aldrich, Cat. No. D9542) in PBT for 3 h at room temperature or overnight at 4 °C. Following four washes with PBT over 1 h, guts were then incubated in Vectashield for at least 30 min before mounting on slides. Phospho-histone H3-positive cells were manually counted in whole midguts using an upright epi-fluorescent Zeiss Axioplan 2 microscope.

**Statistics**. Statistical analysis was performed using Graphpad Prism 7 except for survival curves where OASIS2[58] was used. For all survival curves, *n* numbers, mean and median lifespans and log-rank tests of significance are given in Supplementary Table 1. For other comparisons between two samples, two-tailed Student's *t*-tests were used. For multiple comparisons, one-way analysis of variance with Tukeys test was used. All experiments were performed independently at least twice to confirm reproducibility. Samples were not excluded from the analysis except for Kaplan–Meier survival curves where rare escapers were censored. Samples were not randomised and the investigators were not blinded. The number of samples was determined empirically. All graphs show the mean with error bars of 1 SEM and raw data points are indicated. Statistical significance is indicated by asterisks, where *$p < 0.05$, **$p < 0.01$, ***$p < 0.001$, ****$p < 0.0001$.

**Data availability**. 16S rRNA-sequencing data are deposited at DDBJ with the accession number DRA005828. All relevant data are available from the authors upon request.

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

## Acknowledgements

We acknowledge Marc Dionne, Andrew MacPherson, Naren Srinivasan, Oliver Gordon and members of the Gould lab for critical reading of the manuscript and technical assistance. This work was supported by an Investigator Award to APG from The Wellcome Trust (104566/Z/14/Z) and by the Francis Crick Institute, which receives its core funding from Cancer Research UK (FC001088, the UK Medical Research Council (FC001088), and the Wellcome Trust (FC001088). This work was also supported by grants to F.O. from the Naito Foundation and the Uehara Memorial Foundation.

## Author contributions

F.O. and A.P.G. conceived the project, designed the study and wrote the manuscript. F.O. and C.O.F. performed the experiments and F.O., C.O.F. and A.P.G. analysed the data. All authors edited and approved the final manuscript.

## Additional information

**Competing interests:** The authors declare no competing financial interests.

