## [Peer Review File · Nature Communications]

Reviewers' comments:

Reviewer #1 (Remarks to the Author):

The study investigated developmental effects of the gut microbiome on starvation resistance and lifespan in *Drosophila* fruit flies. The main finding was that depletion of a specific part of the microbiome, *Acetobacter*, increases both phenotypic traits, and that these effects can cross fly generations.

The work opens with a set of experiments where a range of stresses were applied to fly larvae and the effects on adult starvation resistance and longevity measured. Low doses of tBH showed increased lifespan, stored TGs and resistance to starvation. These responses did not appear to be a consequence of a mitohormetic response to oxidative stress. They were correlated with the appearance of rod-shaped microbes in the gut lumen. These could be eliminated with a cocktail of ampicillin, tetracycline and Rifamycin. tBH altered the fly gut microbiome, specifically reducing *Acetobacteriaceae* but not *Lactobacillus* spp.

Main comments

1. The authors state that the fly microbiome shortens adult survival. However, the literature is more mixed on this point than the authors acknowledge, e.g. Heintz & Mair 2014 Cell 156. The fly microbiome is quite labile, and changes with the local environmental microflora. Removing the microbiome is therefore a priori likely to have varying effects depending what is present to start with.
2. The authors mainly used antibiotics to selectively remove parts of the microbiome, in order to deduce the bacterial strains responsible for particular phenotypic consequences for the fly. However, this approach is not state of the art. The Douglas lab have clearly demonstrated, by comparison with the effects of bacterial removal by dechoriation, that antibiotics induce specific side-effects in the fly that have nothing to do with the ablation of the microbiome, which is not surprising given that many of them can induce a long-term impairment of mitochondrial function.
3. A key set of data is reported in Figure 3. However, the approach used in 3a is correlational, based on the use of antibiotic, while the experimental approach was based on infection of adults, while the authors have already shown that manipulation during the larval phase is needed to get the phenotypic effect in adults. These approaches do not directly address the issue of whether specific alterations to the microbiome during development are directly responsible for the adult phenotypes that ensue from treatment with G418 or tBH. There is always the possibility that some other low-abundant bacterium is the culprit, for which I would recommend monoassociation or gnotobiotic lifespans.
4. Several studies have reported phenotypic effects, including persistent ones, of experimental infections of flies with *Acetobacteriaceae* and *Lactobacillus* spp., e.g. PMID: 28724687, PMID: 26439865, PMID: 26439865, PMID: 28062579. It is not clear how the present study advances on these more incisive analyses.
5. The authors suggest that "A key finding is the identification of *Acetobacter* species as a component of the microbiome that limits host lifespan." That is not demonstrated in the paper, and the example of *Lactobacillus plantarum* and *L. brevis* also shows that detrimental or beneficial effects can be species-specific. For that claim, they would have to run lifespans with different *Acetobacter* associations. Given that *A. acetii* might also be much less abundant in other laboratories, this limits the impact of their finding; much more frequently reported are *A. pomorum* and *A. tropicalis*. In addition, I would have liked to see a repetition of the effect with an externally acquired *A. acetii* strain; it might be that they just have an abnormal strain in their fly

stocks.

Specific points:

1. More detail is needed on sequencing methods, read depth and data analysis e.g. the analysis software used. Diversity indices would indicate whether tBH reduces the complexity of the microbiota. An account is needed of the bioinformatic removal of Wolbachia reads in the Canton S strain, which the authors state is Wolbachia positive.
2. The wDah female tBH lifespan is missing. The authors switch between male and female in Fig. 4c+d, without showing the data for the respective other gender, or giving a reason.
3. A lifespan with chronic feeding of *L. plantarum* would help clarify whether increased titer also impacts lifespan.
4. There is no low-dose chronic tBH lifespan showing that the absence of *A. acetii* in adult life only can extend lifespan, only a 20mM lifespan which is toxic, and a 5mM treatment for 6 days after eclosion.
5. I do not see much of a difference for the female starvation assays, which they just claim is 'less pronounced'.
6. Regarding the mechanism, it would have been nice to narrow down the bacterial cue leading to immune activation, for example by feeding with inactivated bacteria, or with bacterial culture supernatant. If *A. pomorum* did not shorten lifespan in their flies, a comparative analysis of *A. pomorum* and *A. acetii* proteome/transcriptome/metabolome/secretome would have been interesting.

Reviewer #2 (Remarks to the Author):

The manuscript describes novel and timely findings that are of high interest to a broad community, namely that: An environmentally-induced change in gut microbiome composition can support long-lasting (and even transgenerational) effects on longevity, triglyceride reserves and starvation tolerance. This work also provides important demonstration of non-beneficial impacts of symbiotic bacteria that were thus far implicated only with positive contributions to their host. While I do have a few questions and suggestions (below), the main conclusions are largely well-supported by the presented evidence and the manuscript is well-written. I therefore recommend publication of a revised manuscript in Nature Communications.

Comments:

- I wonder if the effects on longevity, TAGs and starvation tolerance of the adults are induced solely by *Acetobacter* depletion (i.e. without any contribution from a potential effect of tBH directly on the host). If I am not mistaken, the main causal evidence implicating *Acetobacter* depletion with the induction is the suppression of the (tBH-induced) phenotypes by *Ap* complementation (Fig. 3i,j). The ability to suppress the phenotypes by adding back *Acetobacter* clearly shows that the persistence of the induced phenotypes depends on depletion of *Acetobacter*, but not by itself sufficient to determine that the depletion of *Acetobacter* is also the sole cause of the initial induction. The latter is somewhat supported by the persistence of the phenotypes in (*Acetobacter*-depleted) F1 offspring that were no longer exposed to tBH (Extended Data Fig. 7c,d). However, since we cannot rule out additional transgenerational influence on the host itself, I am still not sure we can completely exclude the possibility that the induction in F0 is due to a combined influence of *Acetobacter* depletion and host-intrinsic effects of tBH. While not obligatory for publication, I would

recommend analyzing the outcome of bacterial removal by dechoriation with and without complementation, but without exposure to tBH (or equivalent stressors) in F0.

Minor comments:

- Regarding the sentence "early life tBH exposure is unlikely to promote longevity via a long-lasting adaptive or mitohormetic response to oxidative stress": I am not convinced that this conclusion is fully supported by the evidence. An alternative scenario could be that the effect of early experience on longevity is mediated by long lasting changes that follow a more transient oxidative stress response (one potential example for these long-lasting changes could be the reported abrogation of age-related ISC overproliferation). I believe that this alternative is also consistent with lack of induction of oxidative stress genes and oxidative stress tolerance in the adult (the latter might seem like a reasonable proxy for longevity, but it is not one and the same as longevity).

- Statement in the abstract: ... "This study identifies a bacterial species in the microbiome that determines host longevity" – I would change "that determines host longevity" to something like "can affect host longevity". Otherwise it gives the impression (which I believe is wrong) that a particular species of bacteria has a general function of determining longevity. Similar reservation applies to statements such as: "A. aceti, is a potent driver of gut IMD hyperactivation and dysfunction during ageing" (middle of last page). The latter gives the impression that all strains of A. aceti (as determined by 16S rRNA) have the same influence in all contexts. The observed functions of a given strain depend on internal and external factors and can change quite rapidly without necessarily being accompanied by variation in 16S rRNA sequence. I would therefore not exclude the possibility that a strain of A. aceti (16S rRNA-wise) will be found to have a different (and potentially even an inverse) influence in other contexts.

On the same ground, I would avoid giving the impression that certain bacteria in complex mammalian microbiomes contribute to age-related diseases and longevity regardless of the type and condition of the host. Same for: "... in the context of a complex microbiome, A. aceti can suppress or outcompete L. plantarum but not vice versa". Suppression by A. aceti in one context does not mean it is expected to hold in general for complex microbiomes (it may not even always hold in cases which consist mainly of Ap and Lp).

- Since the microbiome is typically expected to recover when the environment returns to normal, it might be good to contrast this with the persistence of Acetobacter depletion in F1 following tBH exposure in F0 (especially since the new finding is highly consistent with evidence reported in Fridmann-Sirkis et al.).

- Is the non-monotonic dose response to tBH is also observed in females?

- Extended Data Figure 3 (panels d and e): It would help indicating (in the panels and/or the captions) that the tBH curves correspond to exposure during larval stages.

- Is the higher relative abundance of Lactobacilli in adults vs. larvae (Fig. 2a) also observed in other lines?

Reviewer #3 (Remarks to the Author):

This manuscript addresses how early life exposure to environmental factors such as antibiotics impact long-term health and disease via microbiome alteration, an important and timely issue.

Using *Drosophila* as a model system, the authors report that early life exposure to chemicals, when given during post-embryonic development rather than adulthood, influences adult lifespan

via a microbiome alteration and this effect even persists into the next generation. Importantly this is one of the few studies identifying the causal micro-organism responsible of the host-microbiome interaction phenotype. Using a set of elegant experiments the authors show that the major commensal bacteria species of the *Drosophila* gut, *Acetobacter* sp. are depleted by either the tert-butyl hydroperoxide (tBH) or the antibiotic G418 when given at larval stage. This early treatment reduces age-related innate immune hyperactivation, gut hyperplasia and extend adult lifespan. Importantly, the authors report that reassociating animals treated with the chemical during their young age with a purified culture of *Acetobacter acetii* triggers age-related immune hyperactivation and abrogates the chemical induced lifespan extension.

While the study is robust and well conducted I have few questions related to the experimental setups:

-The lifespan results with 5mM tBH are really different in Fig1e and Fig3J while it seems to be the exact same setup, is this reflecting the variability of the tBH induced lifespan extension? if so, and considering the Fig3J data, the tBH effect can be rather subtle, am I missing something?

-In the light of the weak but significant effect of *Acetobacter acetii* reported in Fig3J I would recommend the authors to test *A.aceti* on the whole range of tBH concentrations used in this study.

-In Fig3c why the authors used 0.5mM tBH while most of the paper reports 5mM tBH effects? Is this a typo?

In addition, this nice phenomenological study would have gained impact if more mechanistical insights were provided such as a clear demonstration that the gut immune hyperactivation by *Acetobacter* sp and gut immune activation dampening by tBH (via loss of *Acetobacter* sp.) is causal to the observed lifespan phenotypes. To address this point the authors could analyse the effect on lifespan of the early tBH treatment in *Imd* gain of function animals (*pirk;pgrp-Ib* double mutant) and in *Imd* loss of function mutant for early tBH treatment followed by association with *A.aceti* at adult stage. Also using gnotobiotic models would be require to compare the tBH impact on lactobacilli-associated flies vs acetobacter-associated flies vs lactobacilli+acetobacter-associated flies, one would expect tBH to be neutral in lactobacilli-associated flies while impactful in acetobacter-associated animals. Finally showing that tBH and G418 have no influence on adult lifespan when given in young germ free animals at the concentration used in this study is vital for the robustness of the demonstration that their effect on lifespan entirely belongs to their impact on the microbiome composition.

Response to Reviewers' comments

Reviewer #1 (Remarks to the Author):

The study investigated developmental effects of the gut microbiome on starvation resistance and lifespan in *Drosophila* fruit flies. The main finding was that depletion of a specific part of the microbiome, *Acetobacter*, increases both phenotypic traits, and that these effects can cross fly generations.

The work opens with a set of experiments where a range of stresses were applied to fly larvae and the effects on adult starvation resistance and longevity measured. Low doses of tBH showed increased lifespan, stored TGs and resistance to starvation. These responses did not appear to be a consequence of a mitohormetic response to oxidative stress. They were correlated with the appearance of rod-shaped microbes in the gut lumen. These could be eliminated with a cocktail of ampicillin, tetracycline and Rifamycin. tBH altered the fly gut microbiome, specifically reducing *Acetobacteriaceae* but not *Lactobacillus* spp.

The Reviewer's summary above quite rightly mentions that the tBH responses (lifespan, TAG and starvation resistance) do not appear to be a consequence of a mitohormetic response. We have now conducted new experiments with a second and more widely used oxidative stressor, paraquat, and this greatly strengthens the general conclusion that oxidants can extend lifespan via microbiome remodelling. Thus, a mitohormetic response of the kind described in the literature (Ristow and Schmeisser, 2014) cannot itself account for the longevity phenotype. Our new experiments now show conclusively that developmental exposure to low dose paraquat (1 mM), like low-dose tBH, can extend lifespan and increase TAG storage (**revised Figure 1f,g**). As with low-dose tBH, this correlates with ablation of *A. aceti* and a concomitant increase in *L. plantarum* (**revised Figure 4c**). Again, as with low dose tBH, reassociation of paraquat flies with a clonal isolate of *A. aceti* (FO1), or a second *Acetobacter* species (FO2), reverts longevity to that of control microbiome flies (**revised Figure 7f**). We conclude that the lifespan extensions we observe in response to low doses of either one of two very different chemical oxidants (paraquat or tBH) require selective depletion of *Acetobacter* from the microbiome.

These new experiments allow us to make the more general and interesting point that early-life exposure to low-dose oxidants can extend *Drosophila* lifespan via remodelling of the microbiome to deplete *Acetobacter* species. This provides an alternative mechanism to mitohormesis for how mild oxidative stress can increase longevity. This has now been emphasized in the rewritten Title, Abstract and Discussion.

Main comments

1. The authors state that the fly microbiome shortens adult survival. However, the literature is more mixed on this point than the authors acknowledge, e.g. Heintz & Mair 2014 Cell 156. The fly microbiome is quite labile, and changes with the local environmental microflora. Removing the microbiome is therefore a priori likely to have varying effects depending what is present to start with.

We acknowledge that the Reviewer is correct about microbiota being variable and the literature reporting both negative and positive effects on longevity. In line with the Reviewer's comments, we now revise the manuscript to be more balanced on this issue and cite the Heintz and Mair Cell paper (Heintz and Mair, 2014). We also cite the two axenic *Drosophila* papers mentioned in this review, one showing beneficial effects of the microbiome upon longevity and the other showing no effect of microbiota upon lifespan (Brummel et al., 2004; Ren et al., 2007).

2. The authors mainly used antibiotics to selectively remove parts of the microbiome, in order to deduce the bacterial strains responsible for particular phenotypic consequences for the fly. However, this approach is not state of the art. The Douglas lab have clearly demonstrated, by comparison with the effects of bacterial removal by dechoriation, that antibiotics induce specific side-effects in the fly that have nothing to do with the ablation of the microbiome, which is not surprising given that many of them can induce a long-term impairment of mitochondrial function.

We acknowledge that antibiotics can induce specific side effects that are not microbiome related. As the Reviewer mentions, this was demonstrated in a paper from the Douglas lab by comparing egg dechoriation versus chlortetracycline treatments (Ridley et al., 2013). However, this is not a valid criticism of our non-antibiotic experimental approaches as specific reassociation of the *Acetobacter*-depleted microbiome of either tBH or paraquat treated animals with a clonal isolate of *A. acetii* (or a second *Acetobacter* species) reverses the beneficial (not harmful) effects on TAG storage and longevity back to control values (**revised figures 7e,f**). This demonstrates that the beneficial effects of low dose tBH and paraquat require their selective anti-bacterial action. We have clarified this important conclusion in the revised Results and Discussion.

Our microbiome complementation assays are state-of-the-art as they provide the best way to test the physiological contribution of one bacterial species (a well characterised clonal isolate) *in the context of a complex microbiome* that is much closer to the complete endogenous microbiome than would be possible using existing dechoriation-based mono-association or gnotobiotic approaches. Also, please see **response to the first Comments of Reviewer #2**, including the limitations of germ-free experiments and the fact that they may expose the host embryo to an oxidant, sodium hypochlorite.

3. A key set of data is reported in Figure 3. However, the approach used in 3a is correlational, based on the use of antibiotic, while the experimental approach was based on infection of adults, while the authors have already shown that manipulation during the larval phase is needed to get the phenotypic effect in adults.

We acknowledge that the G418 antibiotic approach outlined in the original Figure 3a is correlational-this was intentional. The purpose of the G418 experiments is simply to illustrate that a well characterised and widely used antibiotic can mimic the effects of low dose tBH and paraquat (the main focus of our study), in terms of selective *A. acetii* depletion as well as increased TAG storage, and longevity. This strengthens the notion that tBH and paraquat could act via a selective antibiotic mechanism. To identify the underlying mechanism itself, we then focused on tBH (and now also paraquat), not on the use of antibiotics. (*for a reply to the last point about larvae versus adults please see our next response*).

These approaches do not directly address the issue of whether specific alterations to the microbiome during development are directly responsible for the adult phenotypes that ensue from treatment with G418 or tBH.

Our experimental approaches do directly address this issue for treatments with tBH (original manuscript) and now also with paraquat (new experiments in revised manuscript). The key finding here is that re-association with *A. aceti* or another *Acetobacter* species at **adult** stages can revert the TAG storage and longevity phenotypes ensuing from **larval only** exposure to tBH or paraquat. This demonstrates that *Acetobacter* can exert their lifespan-shortening and triglyceride phenotypes by directly acting upon the adult, without acting upon the larva. This does not rule out the existence of all tBH/paraquat effects upon larval host physiology (in fact we provide larval data on changes in gut and whole-body expression of oxidative stress genes). We acknowledge that these conclusions were not very clear in the original manuscript and we have therefore amended the revised Results and, in particular, the revised Discussion.

There is always the possibility that some other low-abundant bacterium is the culprit, for which I would recommend monoassociation or gnotobiotic lifespans.

We refer back to our response to point 2 of this Reviewer for an explanation of why we know that *Acetobacter* species are the culprit. In brief, complementation of oxidant treated microbiomes (selectively depleted of *Acetobacteraceae*) with clonal isolates of *A. aceti* (or new experiments with a second species of *Acetobacter*) specifically reverses the triglyceride storage and longevity phenotypes back to control values. In the revised manuscript, we now demonstrate this longevity reversal for two very different oxidant molecules (tBH and paraquat) and with two different *Acetobacter* species, thus increasing its general relevance.

4. Several studies have reported phenotypic effects, including persistent ones, of experimental infections of flies with *Acetobacteriaceae* and *Lactobacillus* spp., e.g. PMID: 28724687, PMID: 26439865, PMID: 26439865, PMID: 28062579. It is not clear how the present study advances on these more incisive analyses.

We agree with The Reviewer that these three papers are nice incisive analyses of experimental infections of flies with *Acetobacteraceae* and *Lactobacillus* spp. Taking each of these papers in turn, we clarify why our own study provides important new conclusions that constitute a substantial and significant scientific advance:

PMID: 28724687. Morimoto et al. 2017 Biol Lett. doi:10.1098/rsbl.2016.0966.

This short paper contains one figure of data. It reports effects on germ free flies of either sex inoculated with *L. plantarum* or *A. pomorum*. Specifically, it finds that *A. pomorum* in males has a negative effect on mating duration and offspring number and that the presence of this bacterium in both parental sexes gave a very small but significant decrease in their daughters' (but not their sons') body weight. In contrast to our study, the published paper does not provide any link of *A. pomorum* nor any other *Acetobacteraceae* spp. with TAG storage, intestinal function or, most importantly, lifespan.

PMID: 26439865. Erkosar et al. 2015 Cell Host Microbe. doi:10.1016/j.chom.2015.09.001

This interesting paper focuses on a strain of *L. plantarum*. It shows that the previously described ability of this bacterium to enhance larval growth and maturation on a low-protein (yeast) diet involves PGRP-LE/Imd/Relish signaling, which increases intestinal peptidase expression and thus boosts the amino acid levels of the host larva. In contrast to our study, the published paper focuses on

larval growth/maturation not adult phenotypes and does not provide any link of any *Acetobacteraceae* spp. with TAG storage, intestinal function or, most importantly, lifespan.

PMID: 28062579. Tefit and Leulier 2017 J Exp Biol. doi:10.1242/jeb.151522.

This paper again focuses on *L. plantarum*. It shows that larval monoassociation with *L. plantarum* does not alter adult fecundity or fertility, nor does it alter the lifespan of either sex on a standard adult diet. It does, however, moderately increase male but not female lifespan (compared to germ-free controls) on a poor adult diet. In contrast to our study, the published paper uses an artificial microbiome containing only one bacterial species (*L. plantarum*). Importantly, again in contrast to our study, it does not identify any component of the microbiome that is rate limiting for longevity and it does not assess the contribution of any *Acetobacteraceae* spp. to any adult traits nor does it provide any link between *Acetobacter* and TAG storage, intestinal function or lifespan.

5. The authors suggest that "A key finding is the identification of *Acetobacter* species as a component of the microbiome that limits host lifespan." That is not demonstrated in the paper...

We refer the Reviewer back to our detailed response to their points 2 and 3 for why we have now demonstrated this key finding by microbiome complementation with clonal isolates of two different *Acetobacter* species.

..., and the example of *Lactobacillus plantarum* and *L. brevis* also shows that detrimental or beneficial effects can be species-specific. For that claim, they would have to run lifespans with different *Acetobacter* associations. Given that *A. acetii* might also be much less abundant in other laboratories, this limits the impact of their finding; much more frequently reported are *A. pomorum* and *A. tropicalis*. In addition, I would have liked to see a repetition of the effect with an externally acquired *A. acetii* strain; it might be that they just have an abnormal strain in their fly stocks.

We agree with the Reviewer that it is interesting (although not essential to support our original conclusions) to distinguish whether the effects of *A. aceti* upon longevity are species-specific or a more general feature of *Acetobacter*. In line with the Reviewer's comments, we have therefore conducted two new types of experiment to address this:

First, we externally acquired a strain of *A. pomorum* kindly provided by the laboratory of Won Jae-Lee (Shin et al., 2011). Interestingly but frustratingly, this strain of *A. pomorum* failed to colonise our *w^{iso31}* flies in a stable long-term manner and so could not be used for microbiome complementation assays (Reviewers' **Figure R1**).

Figure R1. *A. aceti* FO1 but not *A. pomorum* (from Won Jae-Lee, WJL) colonises tBH-experienced flies in a stable manner. qPCR analysis of male flies exposed to the indicated bacterial species from day 2 to day 8 and then transferred to a control diet until gut dissection at day 16. The bar graphs show the average and SEM with raw data points indicated (N=4). MRS indicates the control of de Man, Rogosa and Sharpe medium with no bacteria.

Second, we isolated a bacterial clone of an additional species of *Acetobacter* present at lower levels than *A. aceti* in our *w iso31* microbiome. We designate this strain as *Acetobacter* FO2. 16S sequencing analysis of FO2 indicates that it is most similar to *A. pomorum* and *A. pasteurianus*, additional HSP60 sequencing analysis gives 99% identity to both species and MALDI-ToF-MS analysis also did not clearly pinpoint FO2 to one of these two species (please see our **revised Methods**). In agreement with the conclusion that multiple *Acetobacter* species are a component of the microbiome that limits host lifespan, we find that add back of either *A. aceti* FO1 or *Acetobacter* FO2 similarly reverses the lifespan extension observed with larval low-dose paraquat treatment (**revised Figure 7f**). This demonstrates directly that at least two different *Acetobacter* species from the indigenous microbiome can limit fly lifespan in a similar manner.

Specific points:

1. More detail is needed on sequencing methods, read depth and data analysis e.g. the analysis software used. Diversity indices would indicate whether tBH reduces the complexity of the microbiota. An account is needed of the bioinformatic removal of *Wolbachia* reads in the Canton S strain, which the authors state is *Wolbachia* positive.

We thank the Reviewer for pointing this out and have now provided an analysis of diversity in the revised **Supplementary Fig. 4** and more details about the analysis methods and *Wolbachia* in the revised **Methods** section. In line with the Reviewer's comments, the rarefaction curves and Shannon entropy suggest a tendency for decreased microbial complexity in tBH-treated guts (revised **Supplementary Figure 4a,b**). Moreover, Bray-Curtis analysis of beta diversity indicates a clear separation between the control and tBH-treated triplicates (revised **Supplementary Fig. 4c**). Regarding *Wolbachia*, we now state in the

Methods that we did not remove the Wolbachia reads from the Canton S data as they amounted to less than 1% of the total sequence reads.

2. The wDah female tBH lifespan is missing. The authors switch between male and female in Fig. 4c+d, without showing the data for the respective other gender, or giving a reason.

We thank the Reviewer for pointing this out and now provide the missing tBH female w^{Dah} survival curve (**Revised Supplementary Figure 1b**).

In line with the Reviewer's comments, we now state in the revised manuscript that intestinal stem cell (ISC) proliferation counts used female flies (original Fig. 4d), as age-related intestinal dysfunction is most prevalent in females(Regan et al., 2016).

3. A lifespan with chronic feeding of *L. plantarum* would help clarify whether increased titer also impacts lifespan.

We agree that this would be an interesting experiment but it is not necessary to support any of the conclusions in our original or revised manuscripts, which focus on the host effects of *Acetobacter* not *Lactobacilli*. Nevertheless, we have shown that tBH experienced flies (*Acetobacter* negative and with an increased titre of *L. plantarum*-see **Revised Figure 3c,3d, 4c**) are slightly longer lived when treated with pan-bacterial antibiotic cocktails (**Revised Figure 6d,e**). This suggests that *L. plantarum* and/or other tBH-resistant bacteria can become mildly limiting for longevity once the major lifespan-shortening component (i.e *Acetobacter*) has been removed.

We have not conducted chronic feeding experiments with multiple inoculations of *L. plantarum*. However, we have data using the same day-2 inoculation protocol that gave stable recolonisation with *A. aceti* FO1 and *Acetobacter* FO2. This clearly shows that inoculation with a clonal isolate (*L. plantarum* FO3) has no significant impact upon longevity (Reviewers' **Figure R2**).

Figure R2. Inoculation with *L. plantarum* FO3 has no significant effect upon the longevity of male w^{iso31} flies in two independent experiments (left and right survival curves).

4. There is no low-dose chronic tBH lifespan showing that the absence of *A. acetii* in adult life only can extend lifespan, only a 20mM lifespan which is toxic, and a 5mM treatment for 6 days after eclosion.

The reason why our manuscript and conclusions focus on larval tBH treatment is that we had found that this gives stable long-term depletion of *Acetobacter*, whereas none of the adult treatments we tried were able to achieve this. For this technical reason, it therefore remains open as to whether or not adult-restricted deletion of *A. acetii* can extend lifespan. Instead we show the converse experiment, namely that adult-restricted addition of *A. acetii* to the microbiome is sufficient to abolish the lifespan extension associated with larval tBH exposure. This clearly demonstrates that *A. acetii* acting upon the host during only adulthood is enough to shorten longevity. Importantly, low-dose chronic tBH lifespans (involving tBH restricted to adulthood) are not necessary to support any of the conclusions in our manuscript, which focuses on the long-term effects of oxidant exposure during development. We have now clarified this issue in the revised Discussion.

5. I do not see much of a difference for the female starvation assays, which they just claim is 'less pronounced'.

We thank the Reviewer for pointing out this lack of clarity. Due to space constraints in the original Nature letter format, we showed data from three different fly strains but only summarised in one sentence that the increase in starvation resistance in tBH flies is "*more robust in males than in females (Extended Data Fig. 2a-f).*" In line with the Reviewer's comments, we have now revised the manuscript to state that the longevity increase is robust between strains in males but strain-dependent in females. e.g a clear difference is visible for females treated with 5mM tBH in the Canton S but not the *w^{iso31}* strain.

6. Regarding the mechanism, it would have been nice to narrow down the bacterial cue leading to immune activation, for example by feeding with inactivated bacteria, or with bacterial culture supernatant.

We thank the Reviewer for their helpful suggested experiments, which would have been nice but are not essential for any of our conclusions. In fact, the suggested experiments would really only be the beginning of a new and long-term study aimed at identifying the bacterial cues from *A. acetii* and other *Acetobacter* and how they activate the immune system in a different way to those from bacteria such as *L. plantarum*. Clearly this lies well beyond the scope of our paper and the first author has recently left the lab.

If *A. pomorum* did not shorten lifespan in their flies, a comparative analysis of *A. pomorum* and *A. acetii* proteome/transcriptome/metabolome/secretome would have been interesting.

We agree that this would have been very interesting, but not anymore as our new experiments now show conclusively that *A. acetii* FO1 and the *A. pomorum/A. pasteurianus* FO2 clones both behave the same way i.e they both shorten lifespan to similar extents (see detailed response to **main comment 5 of this Reviewer**). This strengthens our initial conclusion that members of the *Acetobacter* family other than *A. acetii* can shorten fly lifespan

Reviewer #2 (Remarks to the Author):

The manuscript describes novel and timely findings that are of high interest to a broad community, namely that: An environmentally-induced change in gut microbiome composition can support long-lasting (and even transgenerational) effects on longevity, triglyceride reserves and starvation tolerance. This work also provides important demonstration of non-beneficial impacts of symbiotic bacteria that were thus far implicated only with positive contributions to their host. While I do have a few questions and suggestions (below), the main conclusions are largely well-supported by the presented evidence and the manuscript is well-written. I therefore recommend publication of a revised manuscript in Nature Communications.

We thank the reviewer for acknowledging that our findings are novel and timely, of high interest to a broad community, and for recommending publication of a revised manuscript in Nature Communications.

Comments:

- I wonder if the effects on longevity, TAGs and starvation tolerance of the adults are induced solely by *Acetobacter* depletion (i.e. without any contribution from a potential effect of tBH directly on the host). If I am not mistaken, the main causal evidence implicating *Acetobacter* depletion with the induction is the suppression of the (tBH-induced) phenotypes by Ap complementation (Fig. 3i,j). The ability to suppress the phenotypes by adding back *Acetobacter* clearly shows that the persistence of the induced phenotypes depends on depletion of *Acetobacter*, but not by itself sufficient to determine that the depletion of *Acetobacter* is also the sole cause of the initial induction. The latter is somewhat supported by the persistence of the phenotypes in (*Acetobacter*-depleted) F1 offspring that were no longer exposed to tBH (Extended Data Fig. 7c,d). However, since we cannot rule out additional transgenerational influence on the host itself, I am still not sure we can completely exclude the possibility that the induction in F0 is due to a combined influence of *Acetobacter* depletion and host-intrinsic effects of tBH. While not obligatory for publication, I would recommend analyzing the outcome of bacterial removal by dechoriation with and without complementation, but without exposure to tBH (or equivalent stressors) in F0.

The Reviewer is correct that the *A. aceti* complementation experiments demonstrate that *Acetobacter* depletion during adulthood is necessary for the adult effects upon TAG storage, improved gut function and longevity. This conclusion has further been strengthened in the revised manuscript by a new set of *A. aceti* and *Acetobacter* FO-2 complementation experiments, this time using paraquat (not tBH) to effect *Acetobacter* depletion (**revised Figure 7f, also see response to Reviewer 1**).

The Reviewer also raises an interesting point as to whether the initial induction (but not the persistence) of the phenotypes could involve the combined influence of *Acetobacter* depletion plus host intrinsic effects of tBH. i.e *Acetobacter* depletion in adult flies may not be sufficient by itself to induce the TAG storage, improved gut function and longevity phenotypes. This is an interesting possibility given that we did find that larval tBH exposure activates a local oxidative stress response in the larval gut, albeit one that does not persist into adulthood. However, given that we now show that three different molecules that deplete *Acetobacter* (two pro-oxidants tBH and paraquat, and one antibiotic G418) can all induce similar longevity phenotypes, one would need to hypothesise that they also all induce a similar type of host-intrinsic effect. While this is not impossible, it may be unlikely. The Reviewer suggests experiments (and states they are not obligatory for publication) that are

aimed at eliminating the host-intrinsic response by removing bacteria via dechlorination with sodium hypochlorite. However, these would be subject to the caveat that this chemical is a pro-oxidant that may induce a host oxidative stress response (like tBH and paraquat). Furthermore, monoassociation of dechlorinated germ-free animals with *A. aceti* may not recapitulate the full repertoire of adult phenotypes that we observe using a more natural experimental context, where the control baseline is not germ-free but rather a complex microbiome selectively lacking *Acetobacter*.

In summary, the reviewer is correct that we cannot rule out an additional contribution from host-intrinsic tBH/paraquat/G418 effects and we have now included this interesting point in the revised Discussion.

Minor comments:

- Regarding the sentence “early life tBH exposure is unlikely to promote longevity via a long-lasting adaptive or mitohormetic response to oxidative stress”: I am not convinced that this conclusion is fully supported by the evidence. An alternative scenario could be that the effect of early experience on longevity is mediated by long lasting changes that follow a more transient oxidative stress response (one potential example for these long-lasting changes could be the reported abrogation of age-related ISC overproliferation). I believe that this alternative is also consistent with lack of induction of oxidative stress genes and oxidative stress tolerance in the adult (the latter might seem like a reasonable proxy for longevity, but it is not one and the same as longevity).

We agree with the reviewer and refer back to our response to their preceding and related point. This sentence has now been modified and the Discussion revised accordingly.

- Statement in the abstract: ... “This study identifies a bacterial species in the microbiome that determines host longevity” – I would change “that determines host longevity” to something like “can affect host longevity”. Otherwise it gives the impression (which I believe is wrong) that a particular species of bacteria has a general function of determining longevity. Similar reservation applies to statements such as: “*A. aceti*, is a potent driver of gut IMD hyperactivation and dysfunction during ageing” (middle of last page). The latter gives the impression that all strains of *A. aceti* (as determined by 16S rRNA) have the same influence in all contexts. The observed functions of a given strain depend on internal and external factors and can change quite rapidly without necessarily being accompanied by variation in 16S rRNA sequence. I would therefore not exclude the possibility that a strain of *A. aceti* (16S rRNA-wise) will be found to have a different (and potentially even an inverse) influence in other contexts. On the same ground, I would avoid giving the impression that certain bacteria in complex mammalian microbiomes contribute to age-related diseases and longevity regardless of the type and condition of the host. Same for: “... in the context of a complex microbiome, *A. aceti* can suppress or outcompete *L. plantarum* but not vice versa”. Suppression by *A. aceti* in one context does not mean it is expected to hold in general for complex microbiomes (it may not even always hold in cases which consist mainly of Ap and Lp).

The reviewer makes a very valid point about context and strain dependence and we have modified the revised Abstract and all other text accordingly.

- Since the microbiome is typically expected to recover when the environment returns to normal, it might be good to contrast this with the persistence of *Acetobacter* depletion in F1 following tBH

exposure in F0 (especially since the new finding is highly consistent with evidence reported in Fridmann-Sirkis et al.).

We have now emphasized this point in the revised Discussion and have cited Fridmann-Sirkis et al., 2014 for the G418 microbiome non-recovery and heritability of larval developmental delay (this paper was also cited in the original manuscript as the inspiration for using G418).

- Is the non-monotonic dose response to tBH is also observed in females?

We tested four tBH doses in males (Figure 1 d) but only two doses in females (Figure 1e). Nevertheless, we observed that the tBH-induced increase in female survival is greater at 1mM than at 5mM, at least during the early phase of mortality from the Kaplan-Meier plot. This is broadly similar to what we observed with males and suggests (but does not prove) that both sexes show a non-monotonic response to tBH.

- Extended Data Figure 3 (panels d and e): It would help indicating (in the panels and/or the captions) that the tBH curves correspond to exposure during larval stages.

We have now indicated this on the panels of **revised Figure 2d and 2e**.

- Is the higher relative abundance of Lactobacilli in adults vs. larvae (Fig. 2a) also observed in other lines?

Figure 2a (now revised Figure 3a) shows that the relative abundance of *Lactobacillaceae* in the guts of control (non-tBH treated) w^{iso31} animals is actually **lower** in adults than in larvae. We did not test larval versus adult abundance in Canton S or w^{Dah} so we do not know if this is generalizable to other fly strains.

Reviewer #3 (Remarks to the Author):

This manuscript addresses how early life exposure to environmental factors such as antibiotics impact long-term health and disease via microbiome alteration, an important and timely issue. Using *Drosophila* as a model system, the authors report that early life exposure to chemicals, when given during post-embryonic development rather than adulthood, influences adult lifespan via a microbiome alteration and this effect even persists into the next generation. Importantly this is one of the few studies identifying the causal micro-organism responsible for the host-microbiome interaction phenotype. Using a set of elegant experiments the authors show that the major commensal bacteria species of the *Drosophila* gut, *Acetobacter* sp. are depleted by either the tert-butyl hydroperoxide (tBH) or the antibiotic G418 when given at larval stage. This early treatment reduces age-related innate immune hyperactivation, gut hyperplasia and extends adult lifespan. Importantly, the authors report that reassociating animals treated with the chemical during their young age with a purified culture of *Acetobacter acetii* triggers age-related immune hyperactivation and abrogates the chemical induced lifespan extension.

We thank the reviewer for noting that our manuscript addresses an important and timely issue, uses a set of elegant experiments and, importantly, is one of a few studies that identifies the causal micro-organism responsible for the host-microbiome interaction phenotype.

While the study is robust and well conducted I have few questions related to the experimental setups:
-The lifespan results with 5mM tBH are really different in Fig1e and Fig3J while it seems to be the exact same setup, is this reflecting the variability of the tBH induced lifespan extension? if so, and considering the Fig3J data, the tBH effect can be rather subtle, am I missing something?
-In the light of the weak but significant effect of *Acetobacter acetii* reported in Fig3J I would recommend the authors to test *A. acetii* on the whole range of tBH concentrations used in this study.

We thank the reviewer for acknowledging that our study is robust and well conducted. The Reviewer is correct that the magnitude (days to median and maximum lifespan) and also the precise shape of the tBH and the control survival curves are variable from experiment to experiment (please see **revised Supplementary Table 1**). Such variability is inherent in all published *Drosophila* lifespan studies, not just those involving microbiota, and it is therefore important to compare absolute survival curve parameters within but not between independent experiments. Having said that, our observation that tBH-experienced flies live longer than controls is very reproducible, even if its magnitude varies. Our data show that this observation holds true not only across different experiments but also across three different genetic strains. Furthermore, the revised manuscript now shows that similar lifespan extensions are obtained with three independent methods for selectively depleting *Acetobacter* from the larval microbiome: tBH, G418 and paraquat. The paraquat experiments are new and include longevity reversal via complementation with *A. acetii* FO1 or with *Acetobacter* FO2, which is closest to *A. pomorum* and *A. pasteurianus* (please see **revised Figure 7f** and our response to **Reviewer 1, main comment #5**). The longevity reversals in the paraquat experiments add to, and strengthen, the original tBH data in Fig 3J, referred to by the Reviewer. In conclusion, *Acetobacter* depletion via three different methods is correlated with robust and reproducible lifespan extensions and, for two of these methods, bacterial complementation assays formally demonstrate that *Acetobacter* species are relevant lifespan-shortening agents.

-In Fig3c why the authors used 0.5mM tBH while most of the paper reports 5mM tBH effects? Is this a typo?

This is not a typo. We first used 0.5mM tBH sometime well into our study to identify the lower limit of the concentration that would selectively deplete *A. acetii* and to see if it also gave increased TAG storage and lifespan extension-which it does (**revised Figure 4d-e**). i.e at tBH concentrations from 0.5-5mM, *A. acetii* depletion always goes hand-in-hand with increased TAG storage and extended lifespan.

[REDACTED]

[REDACTED]

[REDACTED]

Finally showing that tBH and G418 have no influence on adult lifespan when given in young germ free animals at the concentration used in this study is vital for the robustness of the demonstration that their effect on lifespan entirely belongs to their impact on the microbiome composition.

It is actually very important *not* to state that the impact of tBH and G418 (and now also paraquat) *entirely* belongs to their impact upon the microbiome. For example, we cannot rule out that oxidants need to act directly on the host, as well as on the microbiome to exert longevity effects. Please see **response to the first Comments of Reviewer #2**, including the limitations of germ-free experiments and the fact that they may expose the host embryo to low levels of an oxidant, sodium hypochlorite. Even if it were somehow possible make larvae germ-free without exposing them to oxidants (even some antibiotics are reported to induce oxidative stress), the reviewer's suggested experiment would not allow us to demonstrate *entire* dependence upon the microbiome. The reason is that it would not rule out a dual mechanism, whereby tBH/G418/paraquat affect lifespan via a direct action upon the larval host *plus* their effect upon *Acetobacter*. If this is the case, the germ-free experiment suggested by the Reviewer would not detect any change in lifespan because the larval host effect is a necessary but not sufficient component of the mechanism that impacts upon lifespan.

In summary, we cannot say that *Acetobacter* depletion is the *entire* mechanism accounting for longevity but we do show very clearly that it is a necessary component. Hence, the bacterial complementation assays demonstrate that the lifespan-extending effects of tBH and paraquat are strictly dependent upon the absence of *Acetobacter* ssp. We acknowledge that the above ideas were not clear in the original manuscript and have now revised the Discussion to leave open the possibility of a dual host-microbiome effect upon longevity.

References

- Brummel, T., Ching, A., Seroude, L., Simon, A.F., and Benzer, S. (2004). *Drosophila* lifespan enhancement by exogenous bacteria. *Proc. Natl. Acad. Sci. U. S. A.* *101*, 12974–12979.
- Fridmann-Sirkis, Y., Stern, S., Elgart, M., Galili, M., Zeisel, A., Shental, N., and Soen, Y. (2014). Delayed development induced by toxicity to the host can be inherited by a bacterial-dependent, transgenerational effect. *Front. Genet.* *5*, 27.
- Guo, L., Karpac, J., Tran, S.L.L., and Jasper, H. (2014). PGRP-SC2 promotes gut immune homeostasis to limit commensal dysbiosis and extend lifespan. *Cell* *156*, 109–122.
- Heintz, C., and Mair, W. (2014). You Are What You Host: Microbiome Modulation of the Aging Process. *Cell* *156*, 408–411.
- Regan, J.C., Khericha, M., Dobson, A.J., Bolukbasi, E., Rattanavirotkul, N., Partridge, L., Oh, B., Ueda, R., Mengin-Lecreux, D., Lemaitre, B., et al. (2016). Sex difference in pathology of the ageing gut mediates the greater response of female lifespan to dietary restriction. *Elife* *5*, e10956.
- Ren, C., Webster, P., Finkel, S.E., and Tower, J. (2007). Increased Internal and External Bacterial Load during *Drosophila* Aging without Life-Span Trade-Off. *Cell Metab.* *6*, 144–152.
- Ridley, E. V, Wong, A.C.N., and Douglas, A.E. (2013). Microbe-dependent and nonspecific effects of procedures to eliminate the resident microbiota from *Drosophila melanogaster*. *Appl. Environ. Microbiol.* *79*, 3209–3214.
- Ristow, M., and Schmeisser, K. (2014). Mitohormesis: Promoting Health and Lifespan by Increased Levels of Reactive Oxygen Species (ROS). *Dose. Response.* *12*, 288–341.
- Shin, S.C., Kim, S.-H., You, H., Kim, B., Kim, A.C., Lee, K.-A., Yoon, J.-H., Ryu, J.-H., Lee, W.-J., Bäckhed, F., et al. (2011). *Drosophila* microbiome modulates host developmental and metabolic homeostasis via insulin signaling. *Science* *334*, 670–674.

REVIEWERS' COMMENTS:

Reviewer #1 (Remarks to the Author):

I am happy with the revisions.

Reviewer #3 (Remarks to the Author):

The authors significantly improved the revised paper and adequately addressed my comments either in the revised manuscript or in their point-by-point letter. I now support the publication of the MS in its current form.

Response to reviewers' comments

Reviewer #1 (Remarks to the Author):

I am happy with the revisions.

We thank the Reviewer for their time and useful input.

Reviewer #3 (Remarks to the Author):

The authors significantly improved the revised paper and adequately addressed my comments either in the revised manuscript or in their point-by-point letter. I now support the publication of the MS in its current form.

We thank the Reviewer for their time and useful input.